# EXPAND AND COMPRESS: EXPLORING TUNING PRINCIPLES FOR CONTINUAL SPATIO-TEMPORAL GRAPH FORECASTING

**Wei Chen, Yuxuan Liang**✉
The Hong Kong University of Science and Technology (Guangzhou)
`onedeanxxx@gmail.com, yuxliang@outlook.com`

## ABSTRACT

The widespread deployment of sensing devices leads to a surge in data for spatio-temporal forecasting applications such as traffic flow, air quality, and wind energy. Although spatio-temporal graph neural networks (STGNNs) have achieved success in modeling various static spatio-temporal forecasting scenarios, real-world spatio-temporal data are typically received in a streaming manner, and the network continuously expands with the installation of new sensors. Thus, spatio-temporal forecasting in streaming scenarios faces dual challenges: the inefficiency of retraining models over newly-arrived data and the detrimental effects of catastrophic forgetting over long-term history. To address these challenges, we propose a novel prompt tuning-based continuous forecasting method, EAC , following two fundamental tuning principles guided by empirical and theoretical analysis: *expand and compress*, which effectively resolve the aforementioned problems with lightweight tuning parameters. Specifically, we integrate the base STGNN with a continuous prompt pool, utilizing stored prompts (*i.e.*, few learnable parameters) in memory, and jointly optimize them with the base STGNN. This method ensures that the model sequentially learns from the spatio-temporal data stream to accomplish tasks for corresponding periods. Extensive experimental results on multiple real-world datasets demonstrate the multi-faceted superiority of EAC over the state-of-the-art baselines, including effectiveness, efficiency, universality, etc. Our code repository is available at https://github.com/Onedean/EAC.

## 1 INTRODUCTION

Spatio-temporal data is ubiquitous in various applications, such as traffic management (Avila & Mezić, 2020), air quality monitoring (Liang et al., 2023), and wind energy deployment (Yang et al., 2024). Spatio-temporal graph neural networks (STGNNs) (Jin et al., 2023; 2024) have become a predominant paradigm for modeling such data, primarily due to their powerful spatio-temporal representation learning capabilities, which consider the both spatial and temporal dimensions of data by learning temporal representations of graph structures. However, most existing works (Li et al., 2017; Wu et al., 2019; Bai et al., 2020; Cini et al., 2023; Han et al., 2024) assume a static setup, where STGNN models are trained on the entire dataset over a limited time period and maintain fixed parameters after training is completed. In contrast, real-world spatio-temporal data (Liu et al., 2024; Yin et al., 2024) typically exists in a streaming format, with the underlying network structure expanding through the installation of new sensors in surrounding areas, resulting in a constantly evolving spatio-temporal network. *Due to computational and storage costs, it is often impractical to store all data and retrain the entire STGNN model from scratch for each time period.*

To address this problem, several straightforward solutions are available, as illustrated in Figure 1. The simplest approach involves ***pre-training*** an STGNN (using node-count-free graph convolution operators) for testing across subsequent periods. However, due to distribution shifts (Wang et al., 2024a), this method often fails to adapt to new period data. Another approach involves model ***retraining*** and prediction on different data windows due to graph expansion. Unfortunately, this neglects the informational gains from historical data, leading to limited performance improvements. To simultaneously resolve the challenges posed by these two issues, a more effective solution is to adopt a continual learning paradigm (Wang et al., 2024d), which is a research area focused on how

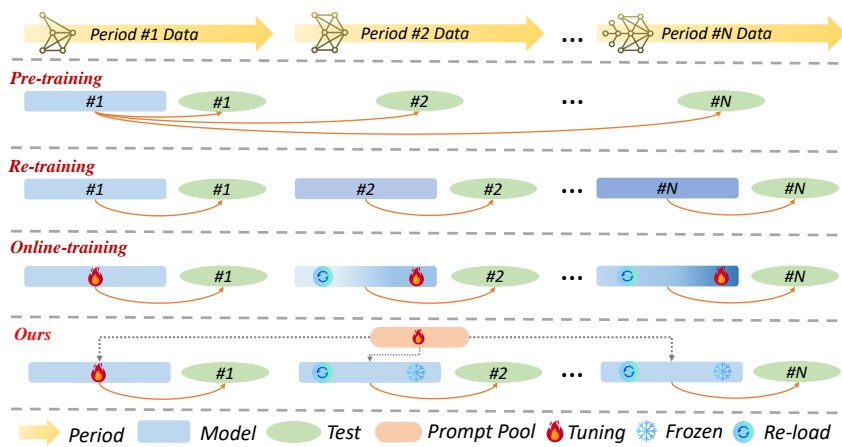

Figure 1: Comparison of classic schemes and `EAC` for continual spatio-temporal forecasting.

systems learn sequentially from continuous related data streams. Specifically, the core idea is to load the previously trained model for new period data and conduct ***online training*** on the current period. *Nevertheless, the notorious problem of catastrophic forgetting (van de Ven et al., 2024) in neural networks often hinders the improvement of online learning performance..*

Current methods for continual spatio-temporal forecasting typically follow various types of continual learning approaches for improvement. For example, TrafficStream (Chen et al., 2021b) comprehensively integrates regularization and replay-based methods to learn and adapt to ongoing data streams while retaining past knowledge to enhance performance. PECPM (Wang et al., 2023b) and STKEC (Wang et al., 2023a) further refine replay strategies to detect stable and changing node data for better adaptation. TFMoE (Lee & Park, 2024) method considers training a sets of mixture of experts models for adapting to new nodes, thereby improving efficiency. *Though promising, the aforementioned methods still involve optimizing the entire STGNN model, resulting in complex tuning costs and failing to mitigate the problem of catastrophic forgetting in a principle way.*

To this end, we propose `EAC` , a novel continual spatio-temporal graph forecasting framework based on ***a continuous prompt parameter pool*** for modeling streaming spatio-temporal data. Specifically, we freeze the base STGNN model during the continual learning process to prevent knowledge forgetting, adapting solely through a dynamically adjustable prompt parameter pool to accommodate the continuously emerging expanded node data while further storing the knowledge acquired from streaming spatio-temporal data. Notably, we explore ***two fundamental tuning principles***, expand and compress, through empirical and theoretical analyses to balance model effectiveness and efficiency. `EAC` has five distinctive characteristics: *(i) **Simplicity***: it accomplishes complex continual learning tasks solely by tuning the prompt parameter pool; *(ii)**Effectiveness*** : it demonstrates consistent performance across multiple real-world datasets; *(iii) **Universality*** : it demonstrates consistent performance across different STGNN architectures; *(iv) **Efficiency***: it accelerates model training speed by effectively freezing the backbone model; and *(v) **Lightweight***: it requires adjustment of only a limited number of parameters in the prompt pool. In summary, our main contributions are:

- We propose a prompt-based continual spatio-temporal forecasting framework `EAC` that is simple, effective, and efficient with lightweight tunable parameters.
- Through empirical observations and theoretical analysis, we explore two tuning principles for continual spatio-temporal forecasting: the heterogeneity property for expansion and low-rank property for compression in our continuous prompt parameter pool.
- Based on the two proposed tuning principles, we introduce two implementation schemes: continuous prompt pool growth and continuous prompt pool reduction.
- Experimental results on different scenarios of real-world datasets from different domains demonstrate the effectiveness and universal superiority of `EAC` .

## 2 RELATED WORK

**Spatio-temporal Forecasting.** Spatio-temporal forecasting originates from time series analysis and can be viewed as a temporal data modeling problem within an underlying network. Traditional statistical models, such as ARIMA (Box & Pierce, 1970) and VAR (Biller & Nelson, 2003), as well

as advanced time series deep learning models (Nie et al., 2023; Wu et al., 2022), can simplify this into a single time series forecasting task. Even though, these methods fail to capture spatio-temporal correlations between different locations, leading to suboptimal performance. STGNNs, due to their inherent ability to aggregate local spatio-temporal information, are considered powerful tools for modeling this data. STGNNs consist of two key components: a graph operator module for spatial relationship modeling, typically categorized into spectral GNNs (Yu et al., 2017), spatial GNNs (Li et al., 2017), or hybrids, and a sequence operator module for temporal relationship modeling, which can be recurrent-based (Pan et al., 2019), convolution-based (Wu et al., 2019), attention-based (Guo et al., 2019), or a combination of these networks. *However, most spatio-temporal graph forecasting models focus on static settings with limited-period forecasting scenarios.*

**Continual Learning.** Continual learning is a technique for sequentially training models as data from related tasks arrives in a streaming manner. Common approaches include regularization-based (Kirkpatrick et al., 2017), replay-based (Rolnick et al., 2019), and prototype-based (De Lange & Tuytelaars, 2021) methods, all aimed at learning knowledge from new tasks while retaining knowledge from previously tasks. However, these methods primarily focus on vision and text domains (Wang et al., 2024d), assuming that samples are *i.i.d.* TrafficStream (Chen et al., 2021b) first integrates the ideas of regularization and replay into continual spatio-temporal forecasting scenarios. PECPM (Wang et al., 2023b) and STKEC (Wang et al., 2023a) further incorporate prototype-based ideas for enhancement. TFMoE (Lee & Park, 2024) advances replay data into a generative reconstruction approach, equipped with a mixture of experts model. Additionally, some methods consider diverse perspectives such as few-shot scenarios (Wang et al., 2024b), large-scale contexts (Wang et al., 2024c), and combinations with reinforcement learning (Xiao et al., 2022) and data augmentation (Miao et al., 2024). *Nonetheless, most methods in dynamic scenarios still require tuning all STGNN parameters, resulting in the dual challenges of catastrophic forgetting and inefficiency.*

**Prompt Learning.** Prompt learning suggest simply tuning frozen language or vision models to perform downstream tasks by learning prompt parameters attached to the input to guide model predictions. Some studies (Yuan et al., 2024; Li et al., 2024) attempt to integrate it into spatio-temporal forecasting, but they still focus on static scenarios. Other methods have applied it to continual learning contexts; however, they only concentrate on vision (Wang et al., 2022) and text (Razdaibiedina et al., 2023) domains. In contrast to the various prompt-based approaches in the existing literature, a naive application of prompt learning in our context is to append learnable parameters $P$ (referred to as prompts) to the original spatio-temporal data $X$, resulting in a fused embedding $X' = [X \parallel P]$, which is then fed into the base STGNN model $f_\theta(X')$ for spatio-temporal forecasting. *Notably, we design a novel prompt pool learning mechanism, guided by two tuning principles derived from empirical and theoretical analysis, to model continual spatio-temporal forecasting.*

## 3 PRELIMINARIES

**Definition (Dynamic Streaming Spatio-temporal Graph).** We consider a dynamic streaming spatio-temporal graph $\mathbb{G} = (\mathcal{G}_1, \mathcal{G}_2, \ldots, \mathcal{G}_\mathcal{T})$, for every time interval $\tau$, the network dynamically grows, *i.e.*, $\mathcal{G}_\tau = \mathcal{G}_{\tau-1} + \Delta\mathcal{G}_\tau$. Specifically, the network in the $\tau$-th time interval is modeled by the graph $\mathcal{G}_\tau = (\mathcal{V}_\tau, \mathcal{E}_\tau, A_\tau)$, where $\mathcal{V}_\tau$ is the set of nodes corresponding to the $|\mathcal{V}_\tau| = n$ sensors in the network, and $\mathcal{E}_\tau$ signifies the edges connecting the node set, which can be further represented by the adjacency matrix $A_\tau \in \mathbb{R}^{n_\tau \times n_\tau}$. The node features are represented by a three-dimensional tensor $X_\tau \in \mathbb{R}^{n \times t \times c}$, denoting the $c$ features of the records of all $n$ nodes observed on the graph $\mathcal{G}_\tau$ in the past $t$ time steps. Following (Chen et al., 2021b), $c$ here is usually only a numerical value.

**Problem (Continual Spatio-temporal Graph Forecasting).** The continual spatio-temporal graph forecasting can be viewed as learning the optimal prediction model for the current stage from dynamic streaming spatio-temporal graph data. Specifically, given the training data $\mathcal{D} = \{D_\tau | (\mathcal{G}_\tau, X_\tau, Y_\tau)\}_{\tau=1}^{\mathcal{T}} \sim \mathcal{P}$ from a sequence of streaming data, our goal is to incrementally learn the optimal model parameters $f_{\theta^*}$ from the sequential training set. For the current $\tau$-th time interval, the model is optimized to minimize:

$$f_{\theta^{(\tau)*}} = \operatorname*{argmin}_{\theta^{(\tau)}} \mathbb{E}_{D_\tau \sim \mathcal{P}^{(\tau)}}[\mathcal{L}(f_{\theta^{(\tau)}}(\mathcal{G}_\tau, X_\tau), Y_\tau)], \tag{1}$$

where $f_{\theta^{(\tau)*}}$ represents the optimal model that achieves the minimum loss when trained on the data from the current period $\tau$. The loss function $\mathcal{L}(\cdot)$ measures the discrepancy between the predicted signals $\widehat{Y}_\tau = f_{\theta^{(\tau)*}}(\mathcal{G}_\tau, X_\tau) \in \mathbb{R}^{n \times t \times c}$ for next $t$ time steps and the ground-truth $Y_\tau$.

## 4 METHODOLOGY

In this section, we propose two tuning principles, ***expand*** and ***compress***, through detailed empirical observations and theoretical analysis. Based on these principles, we apply them to a prompt parameter pool to develop the continual spatio-temporal graph forecasting framework EAC (as shown in Figure 2). Specifically, we design a node-level prompt parameter pool corresponding to the input spatio-temporal data of different nodes, jointly optimized within the STGNN backbone. ❶ For the expand process, empirical studies reveal that the prompt parameter pool adapts to dynamic heterogeneity, which we further analyze theoretically. Building on this, we show that expanding prompt parameters for newly introduced nodes effectively accommodates heterogeneity in continuous spatio-temporal scenarios. ❷ For the compress process, empirical results indicate that the prompt parameter pool exhibits a low-rank property, which we formalize through detailed analysis. Based on this, we show that high-dimensional prompt parameters can be compressed into two low-dimensional components, mitigating parameter inflation caused by expansion in continuous spatio-temporal scenarios. We also summarize the workflow of EAC in Algorithm 1, and provide a detailed explanation of the continual learning process in Appendix B.

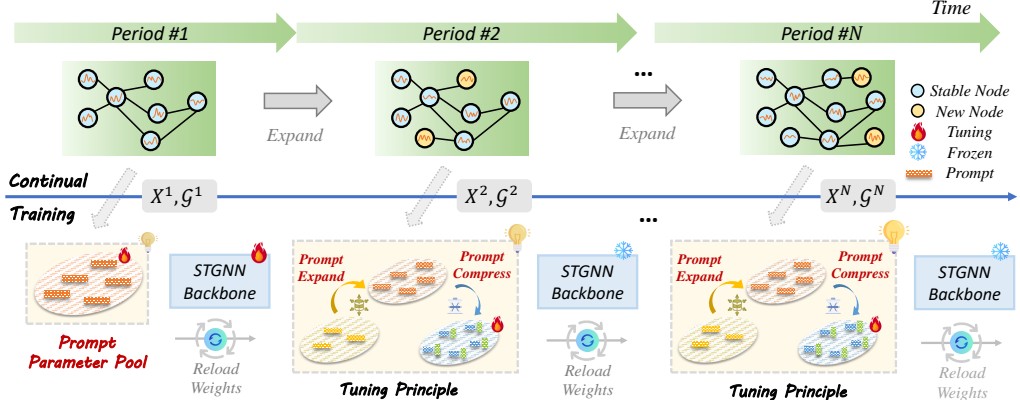

Figure 2: The overall architecture of our proposed EAC .

### 4.1 EXPAND: HETEROGENEITY-GUIDED CONTINUOUS PROMPT POOL GROWTH

**Insight.** As mentioned above, fine-tuning an existing STGNN model with new data streams often leads to catastrophic forgetting (van de Ven et al., 2024). While previous methods have proposed some mitigative strategies (Chen et al., 2021b; Wang et al., 2023a;b), these solutions are not entirely avoidable. *A straightforward solution is to isolate parameters, freeze the old model, and dynamically adjust the network structure to incorporate adaptable learning parameters*. Recently, there has been an increasingly common consensus in spatio-temporal forecasting to introduce node-specific trainable parameters as spatial identifiers to achieve higher performance (Shao et al., 2022; Liu et al., 2023; Dong et al., 2024; Yeh et al., 2024). Although some empirical evidence (Shao et al., 2023; Cini et al., 2024) supports their predictive performance in static scenarios, there has been no root analysis to explain why they are useful, when they are applicable, and in what contexts they are most suitable. However, we find this closely aligns with our motivation and extend it to the continual spatio-temporal forecasting setting by providing a reasonable explanation from the perspective of heterogeneity to address these questions. Specifically, spatio-temporal data generally exhibit two characteristics: *correlation* and *heterogeneity* (Geetha et al., 2008; Wang et al., 2020). The former is naturally captured by various STGNNs, as they automatically aggregate local spatial and temporal information. However, given the message-passing mechanism of STGNNs, the latter is clearly not captured. Therefore, we argue that the introduction of node prompt parameter pool likely enhances the model's ability to capture heterogeneity by expanding the expressiveness of the feature space.

**Empirical Observation.** To quantitatively analyze heterogeneity, we consider the dispersion of node feature vectors in the feature space (Fan et al., 2024). We first define the *Average Node Deviation* $(D(\cdot))$ metric as: $D(X) = \frac{1}{n \times n} \sum_{i=1}^{n} \sum_{j=1}^{n} \sum_{k=1}^{d} (X_{ik} - X_{jk})^2$, where $X \in \mathbb{R}^{n \times d}$ represent the feature matrix composed of $n$ node vectors, each with $d$ dimensions. This metric quantifies the degree of dispersion between pairs of node vectors within the feature matrix, reflecting the ability to express heterogeneity. We use this indicator to plot the dispersion degree of the feature matrix

for the pems-stream dataset across different periods as node prompt parameters are injected during the learning process. As shown in Figure 3, two phenomena are clearly observed: ❶ Within the same period, the dispersion of the node feature space continuously expands throughout the learning process, reflecting the increasing ability of prompt parameters to represent heterogeneity. ❷ Across different periods, the dispersion of the feature space in the current period expands further compared to the previous period, showing the continuous capture ability of the prompt parameter for heterogeneity.

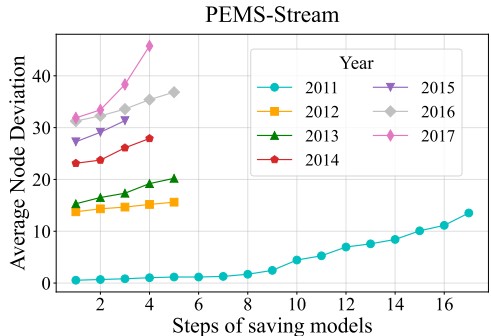

Figure 3: Heterogeneity measurement.

**Theoretical Analysis.** Below, we provide a theoretical analysis of the above empirical results.

**Proposition 1.** *For an original node input feature matrix $X = [x_1, \cdots, x_n] \in \mathbb{R}^{n \times d}$, we introduce a node prompt parameter matrix $P = [p_1, \cdots, p_n] \in \mathbb{R}^{n \times d}$. Through a spatio-temporal learning function $f_\theta$ with invariance, a new feature matrix $X^\theta = f(\theta; X, P)$ is obtained, satisfying:*

$$D(X^\theta) - D(X) = 2\left(\frac{1}{n}\sum_{i=1}^{n}\|p_i^\theta\|^2 - \|\mu_p^\theta\|^2\right) \geq 0, \tag{2}$$

*where $P^\theta = [p_1^\theta, \cdots, p_n^\theta] \in \mathbb{R}^{n \times d}$ represents the optimized prompt parameter matrix, and $\mu_p^\theta = \frac{1}{n}\sum_{i=1}^{n} p_i^\theta$ is the mean vector of the parameter matrix.*

*Proof.* For more details, refer to the supplementary materials in appendix A.1. □

> ***Tuning Principle I:*** *Prompt Parameter Pool Can Continuously Adapt to Heterogeneity Property.*

**Implementation Details.** Based on the above analysis, we present the implementation details for expand process in continuous spatio-temporal forecasting scenarios. Specifically, we continuously maintain a prompt parameter pool $\mathcal{P}$. For the initial static stage, we provide each node with a learnable parameter vector, and the matrix $P^{(1)}$ of all such vectors is added to the parameter pool $\mathcal{P} = [P^{(1)}]$. Follow the Occam's razor, we adopt a simple yet effective fusion method, where the prompt pool and the corresponding input node features are added element-wise. The prompt parameter pool $\mathcal{P}$ is then trained together with the base STGNN model. For subsequent period $\tau$, we only provide prompt parameter vectors for newly added nodes, and the resulting matrix $P^{(\tau)}$ is added to the prompt pool $\mathcal{P} = [P^{(1)}, P^{(2)}, \cdots, P^{(\tau-1)}]$. As we analyzed, we *freeze* the STGNN backbone and *only tuning the prompt pool* $\mathcal{P}$, effectively reducing computational costs and accelerate training.

## 4.2 COMPRESS: LOW-RANK-GUIDED CONTINUOUS PROMPT POOL REDUCTION

**Insight.** While node-customized prompt parameter pools are highly effective, an unavoidable challenge arises in our scenario of continuous spatio-temporal forecasting: the number of prompt parameters continuously increases with the addition of new nodes across consecutive periods, leading to parameter inflation. Despite the existence of numerous well-established studies that enhance the efficiency of spatio-temporal prediction (Bahadori et al., 2014; Yu et al., 2015; Chen et al., 2023; Ruan et al., 2024) and imputation (Chen et al., 2020; 2021a; Nie et al., 2024) tasks using techniques such as compressed sensing and matrix / tensor decomposition, these study typically focus solely on the original spatio-temporal data. *An intuitive solution is to similarly apply low-rank matrix approximations to the prompt learning parameter pool, thereby reducing the number of learnable parameters while maintaining performance.* However, for the prompt learning parameter pool, it remains to be validated whether it exhibits redundancy characteristics akin to spatio-temporal data and whether these properties hold in the continuous spatio-temporal forecasting setting.

**Empirical Observation.** To explore redundancy, we conduct a spectral analysis of the prompt parameter pool $\mathcal{P}$. Specifically, for the models optimized annually on the PEMS-Stream dataset, we first apply singular value decomposition to the extended prompt parameter pool introduced in the

previous section and plot the normalized cumulative singular values for different years, as shown in Figure 4 left. It can be observed that ❶ all years exhibit a clear long-tail spectral distribution, indicating that most information from the parameter matrix $\mathcal{P}$ can be recovered from the first few largest singular values. In Figure 4 right, we also present a heatmap of the normalized cumulative singular values at the sixth largest singular value for different years at different steps, revealing that, ❷ despite some variations across years, the overall processes for all years maintain a high concentration of information ($> 0.75$), suggesting a low-rank property for $\mathcal{P}$.

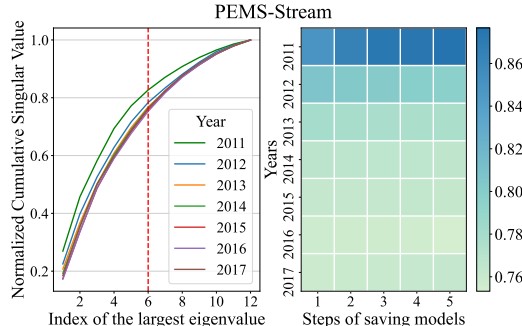

Figure 4: Low-rank measurement.

**Theoretical Analysis.** Below, we provide a theoretical analysis of the above empirical results.

**Proposition 2.** *Given the node prompt parameter matrix $P \in \mathbb{R}^{n \times d}$, there will always be two matrices $A \in \mathbb{R}^{n \times k}$ and $B \in \mathbb{R}^{k \times d}$ such that $P$ can be approximated as $AB$ when the nodes $n$ grow large, and satisfy the following probability inequality:*

$$\Pr\left(\|P - AB\|_F \le \epsilon\|P\|_F\right) \ge 1 - o(1) \ \text{ and } \ k = \mathcal{O}\left(\log(\min(n, d))\right)$$

*where $o(1)$ represents a term that becomes negligible even as $n$ grows large.*

*Proof.* For more details, refer to the supplementary materials in appendix A.2. □

> **Tuning Principle II:** *Prompt Parameter Pool Can Continuously Satisfy the Low-rank Property.*

**Implementation Details.** Formally, we present the implementation details for compress process in continuous spatio-temporal forecasting scenarios based on the aforementioned analysis. Specifically, for the initial static stage, we approximate the original prompt parameter $P^{(1)}$ using the product of the subspace parameter matrix $A^{(1)}$ and the adjustment parameter matrix $B$. For subsequent periods $\tau$, we provide only the subspace parameter matrix $A^{(\tau)}$ for the newly added node vectors, approximating the prompt parameter $P^{(\tau)}$ through the product with the adjustment parameter matrix $B$. As analyzed, the *dimensionality of the subspace parameter* matrix $A$ is *significantly smaller* than that of the *prompt parameter $P$*, while the number of parameters in the adjustment matrix $B$ remains constant; thus, we effectively mitigate the inflation issue.

## 5 EXPERIMENTS

In this section, we conduct extensive experiments to investigate the following research questions:

- **RQ1:** Can EAC outperform previous methods in accuracy across various tasks? *(Effectiveness)*
- **RQ2:** Can EAC have a consistent improvement on various types of STGNNs? *(Universality)*
- **RQ3:** How efficient is EAC compared to different methods during the training phase? *(Efficiency)*
- **RQ4:** How many parameters does EAC require tuning compared to baselines? *(Lightweight)*
- **RQ5:** How does EAC compare to other common prompt-adaptive learning method? *(Simplicity)*

### 5.1 EXPERIMENTAL SETUP

**Dataset and Evaluation Protocol.** We use real-world spatio-temporal graph datasets from three domains: transportation, weather, and energy, encompassing common streaming spatio-temporal forecasting scenarios. The transportation dataset, ***PEMS-Stream***, is derived from benchmark datasets in previous research (Chen et al., 2021b), covering dynamic traffic flow in Northern California from 2011 to 2017 across seven periods. The weather dataset, ***Air-Stream***, originates from the real-time urban air quality platform of the Chinese Environmental Monitoring Center [1], capturing dynamic air quality indicators at monitoring stations across various regions of China from 2016 to 2019 over

---

[1]https://air.cnemc.cn:18007/

four periods. The energy dataset, ***Energy-Stream***, comes from a spatial dynamic wind power fore-casting dataset provided by a power company during the KDD Cup competition (Zhou et al., 2022), containing monitoring metrics for wind farms over a span of 245 days across four periods. During the experiments, each dataset's temporal dimension is split into training, validation, and test sets in a 6:2:2 ratio for each period, employing an early stopping mechanism. According to the established protocol (Chen et al., 2021b), we use the past 12 steps to predict the next 3, 6, 12 steps, and the mean value. Evaluation metrics include *Mean Absolute Error (MAE)*, *Root Mean Square Error (RMSE)*, and *Mean Absolute Percentage Error (MAPE)* averaged over all periods. Further details regarding the datasets and evaluation metrics are available in Appendix C.1.

**Baseline and Parameter Setting.** Following the default settings (Chen et al., 2021b; Wang et al., 2023a; Lee & Park, 2024), we employ the same STGNN as the backbone network and consider the following baseline methods for comparison:

- ***Pretrain-ST***: For each dataset, we train the STGNN backbone using only the spatio-temporal graph data from the first period and directly use this network to predict results on the test sets in subsequent periods.
- ***Retrain-ST***: For each dataset, we train a new backbone network for the spatio-temporal graph data of each period and use the corresponding network to predict results on the test set of the current.
- *Online-ST*: For each dataset, we iteratively train a backbone network in an online manner, where the model weights from the previous period serve as initialization for the current period.
    - ‡ ***Online-ST-AN***: Train on the complete node data of the current period's spatio-temporal graph, tuning the entire model with the model trained from the previous year as initialization.
    - ‡ ***Online-ST-NN***: Train on the newly added node data of the current period's spatio-temporal graph, tuning the entire model with the model trained from the previous year as initialization. ***TFMoE*** (Lee & Park, 2024) improved this using a mixture of expert models technique.
    - ‡ *Online-ST-MN*: Train on the mixed node data (new nodes + some old nodes) of the current period's spatio-temporal graph, tuning the entire model with the model trained from the previous year as initialization. Existing methods typically focus on this aspect, including ***Traffic-Stream*** (Chen et al., 2021b), ***PECPM*** (Wang et al., 2023b), and ***STKEC*** (Wang et al., 2023a).

For both the baseline method and our method, we set parameters uniformly according to the recommendations in previous paper (Chen et al., 2021b) to ensure fair comparison. Our only hyper-parameter $k$ is set to 6. We repeated each experiment five times and report the mean and standard deviation (indicated by gray $\pm$) of all methods. More details about baseline settings, see Appendix C.2.

## 5.2 EFFECTIVENESS STUDY (RQ1)

**Overall Performance.** We report a comparison between EAC and typical schemes (including representative improved methods [2]) in Table 1, where the best results are highlighted in **bold pink** and the second-best results in underlined blue. $\Delta$ indicates the reduction of MAE compared to the second-best result, or the increase of other results relative to the second best result. Moreover, due to the unavailability of official source code for *PECMP* and *TFMoE*, along with the more complex backbone network used by *TFMoE*, the comparisons may be unfair. Nonetheless, we also include comparable reported values, as shown in Table 2. Based on the results, we observe the following:

❶ *Pretrain-ST* methods generally yield the poorest results, especially on smaller datasets (*i.e.*, *Engery-Stram*), aligning with the intuition that they directly use a pre-trained model for zero-shot forecasting in subsequent periods. Even with better pre-training on larger dataset (*i.e.*, *Air-Stream*), performance remains mediocre. ❷ *Retrain-ST* methods also exhibit unsatisfactory results, as they rely on limited data to train specific phase models without effectively utilizing historical information gained from the pretrained model. ❸ *Online-ST-NN* methods perform poorly, as they fine-tune the pretrained model using only new node data differing from the old pattern. Despite *TF-MoE*'s improvements through complex design, severe catastrophic forgetting remains an issue. ❹ *Online-ST-MN* methods strike a balance between performance and efficiency, showing some improvements, particularly on small datasets (*e.g.*, *STKEC* in *Energy-Stream*), due to limited node pattern memory. ❺ *Online-ST-AN* methods typically achieve suboptimal results, as they fine-tune the pretrained model on the full data across different periods, approximating the performance boundary

---

[2]Notably, the core code of *STKEC* is not available, so we carefully reproduce and report average results.

Table 1: Comparison of the overall performance of the classical scheme and `EAC` .

| Datasets | | Air-Stream (1087 → ⋯ → 1202) | | | | PEMS-Stream (655 → ⋯ → 871) | | | | Energy-Stream (103 → ⋯ → 134) | | | |
|---|---|---|---|---|---|---|---|---|---|---|---|---|---|
| Method | Metric | 3 | 6 | 12 | Avg. | 3 | 6 | 12 | Avg. | 3 | 6 | 12 | Avg. |
| **Retrain-ST** | MAE | 18.59±0.39 | 21.53±0.29 | 24.83±0.26 | 21.33±0.29 | 12.96±0.14 | 14.06±0.10 | 16.36±0.11 | 14.24±0.12 | 5.56±0.14 | 5.46±0.12 | 5.45±0.09 | 5.48±0.12 |
| | RMSE | 29.20±0.70 | 34.31±0.59 | 39.61±0.57 | 33.77±0.62 | 20.88±0.17 | 22.96±0.15 | 26.95±0.19 | 23.20±0.16 | 5.75±0.12 | 5.70±0.11 | 5.80±0.09 | 5.72±0.11 |
| | MAPE (%) | 23.72±0.88 | 27.67±0.68 | 32.50±0.46 | 27.54±0.66 | 18.51±0.61 | 19.98±0.42 | 23.31±0.24 | 20.30±0.44 | 54.35±2.11 | 54.61±2.08 | 55.60±1.55 | 54.74±2.06 |
| | Δ | + 1.58% | + 1.03% | + 0.52% | + 0.99% | + 1.25% | + 1.00% | + 1.17% | + 1.13% | + 4.70% | + 2.24% | + 0.73% | + 2.23% |
| **Pretrain-ST** | MAE | 19.58±0.20 | 22.72±0.16 | 26.00±0.23 | 22.44±0.20 | 14.13±0.28 | 15.17±0.26 | 17.35±0.29 | 15.33±0.27 | 10.65±0.00 | 10.66±0.02 | 17.10±6.42 | 17.05±6.39 |
| | RMSE | 31.46±0.20 | 36.78±0.16 | 41.96±0.23 | 36.15±0.34 | 21.77±0.25 | 23.79±0.26 | 27.73±0.35 | 24.04±0.27 | 10.88±0.12 | 10.92±0.13 | 11.02±0.15 | 10.93±0.13 |
| | MAPE (%) | 24.05±1.12 | 28.46±1.23 | 33.48±1.13 | 28.16±1.17 | 30.86±3.34 | 32.07±3.04 | 34.45±3.24 | 32.20±3.17 | 171.88±3.79 | 172.77±4.25 | 174.07±4.81 | 172.71±4.12 |
| | Δ | + 6.99% | + 6.61% | + 5.26% | + 6.25% | + 10.39% | + 8.97% | + 7.29% | + 8.87% | + 100.56% | + 99.62% | + 216.08% | + 218.09% |
| **Online-ST-AN** | MAE | 18.30±0.55 | 21.31±0.49 | 24.70±0.49 | 21.12±0.51 | 12.80±0.06 | 13.92±0.05 | 16.17±0.10 | 14.08±0.05 | 5.47±0.08 | 5.46±0.09 | 5.47±0.11 | 5.47±0.08 |
| | RMSE | 28.54±0.64 | 33.87±0.63 | 39.33±0.68 | 33.28±0.64 | 20.66±0.06 | 22.73±0.06 | 26.64±0.15 | 22.96±0.06 | 5.62±0.05 | 5.66±0.06 | 5.76±0.07 | 5.67±0.05 |
| | MAPE (%) | 23.43±0.81 | 27.43±0.63 | 32.39±0.50 | 27.34±0.66 | 17.86±0.59 | 19.37±0.70 | 22.92±1.05 | 19.73±0.74 | 52.70±1.34 | 53.25±1.54 | 54.50±1.71 | 53.36±1.51 |
| | Δ | – | – | – | – | – | – | – | – | + 3.01% | + 2.24% | + 1.10% | + 2.05% |
| **Online-ST-NN** | MAE | 19.38±1.97 | 22.24±1.81 | 25.50±1.65 | 22.05±1.83 | 14.68±0.91 | 16.57±1.25 | 20.64±2.25 | 16.95±1.39 | 5.51±0.05 | 5.50±0.05 | 5.49±0.07 | 5.50±0.05 |
| | RMSE | 29.57±2.23 | 34.49±1.67 | 39.62±1.24 | 33.97±1.78 | 24.30±2.00 | 28.21±3.09 | 36.77±6.60 | 29.05±3.63 | 5.65±0.04 | 5.68±0.05 | 5.76±0.06 | 5.70±0.04 |
| | MAPE (%) | 23.99±2.55 | 28.02±2.08 | 33.17±1.64 | 27.96±2.13 | 18.93±0.57 | 20.69±0.40 | 24.89±0.62 | 21.15±0.45 | 54.98±1.67 | 55.10±1.11 | 55.62±0.37 | 55.17±1.08 |
| | Δ | + 5.90% | + 4.36% | + 3.23% | + 4.40% | + 14.68% | + 19.03% | + 27.64% | + 20.38% | + 3.76% | + 2.99% | + 1.47% | + 2.61% |
| **TrafficStream** | MAE | 18.66±1.21 | 21.59±0.99 | 24.90±0.79 | 21.39±1.02 | 12.89±0.05 | 14.03±0.11 | 16.39±0.04 | 14.22±0.13 | 5.58±0.05 | 5.57±0.05 | 5.58±0.07 | 5.57±0.05 |
| | RMSE | 29.06±1.64 | 34.22±1.27 | 39.54±1.00 | 33.65±1.35 | 20.78±0.13 | 22.90±0.25 | 26.98±0.53 | 23.16±0.27 | 5.73±0.06 | 5.76±0.06 | 5.86±0.07 | 5.77±0.06 |
| | MAPE (%) | 24.23±1.86 | 28.12±1.50 | 32.99±1.13 | 28.03±1.52 | 17.86±0.41 | 19.50±0.86 | 23.43±2.25 | 19.95±1.02 | 53.87±1.99 | 54.06±1.24 | 54.92±0.71 | 54.16±1.21 |
| | Δ | + 1.96% | + 1.31% | + 0.80% | + 1.27% | + 0.70% | + 0.79% | + 1.36% | + 0.99% | + 5.08% | + 4.30% | + 3.14% | + 3.91% |
| **STKEC** | MAE | 19.42±1.27 | 22.24±1.17 | 25.44±1.05 | 22.06±1.20 | 12.85±0.05 | 13.98±0.04 | 16.25±0.04 | 14.14±0.04 | 5.31±0.27 | 5.34±0.25 | 5.41±0.15 | 5.36±0.22 |
| | RMSE | 30.28±1.65 | 35.09±1.36 | 40.11±1.13 | 34.61±1.43 | 20.73±0.09 | 22.81±0.08 | 26.73±0.07 | 23.04±0.07 | 5.50±0.19 | 5.56±0.20 | 5.72±0.10 | 5.59±0.17 |
| | MAPE (%) | 25.21±2.69 | 28.83±2.22 | 33.30±1.64 | 28.71±2.23 | 17.87±0.14 | 19.25±0.17 | 22.33±0.16 | 19.53±0.15 | 50.10±1.67 | 50.93±1.51 | 52.40±1.10 | 51.04±1.44 |
| | Δ | + 6.12% | + 4.36% | + 2.99% | + 4.45% | + 0.39% | + 0.43% | + 0.49% | + 0.42% | – | – | – | – |
| **EAC** | MAE | **18.11±0.27** | **20.87±0.17** | **24.15±0.14** | **20.75±0.20** | **12.65±0.03** | **13.45±0.05** | **14.92±0.11** | **13.53±0.06** | **5.08±0.10** | **5.09±0.10** | **5.15±0.10** | **5.10±0.10** |
| | RMSE | **27.78±0.47** | **32.88±0.42** | **38.22±0.31** | **32.35±0.40** | **20.24±0.06** | **21.86±0.09** | **24.17±0.17** | **21.77±0.10** | **5.46±0.07** | **5.46±0.09** | **5.46±0.10** | **5.33±0.10** |
| | MAPE | **23.12±0.20** | **26.91±0.07** | **31.79±0.05** | **26.89±0.12** | **17.80±0.08** | **18.79±0.08** | **20.82±0.16** | **18.98±0.08** | **47.53±2.71** | **48.20±2.68** | **50.55±2.60** | **48.56±2.67** |
| | Δ | - 1.03% | - 2.06% | - 2.26% | - 1.75% | - 1.17% | - 3.33% | - 7.73% | - 3.90% | - 4.33% | - 4.68% | - 4.80% | - 4.85% |

of continual-based methods while still suffering some knowledge forgetting. ❻ Our `EAC` consistently improves all metrics across all types of datasets. We attribute this to its ability to continuously adapt to the complex information and knowledge forgetting challenges inherent in continual spatio-temporal learning scenarios by tuning the prompt pool with heterogeneity-capturing parameters.

Table 2: Performance comparison of the improved method with `EAC` on *PEMS-Stream* benchmark.

| Model | Avg. @ MAE | Avg. @ RMSE | Simplicity | Lightweight |
|---|---|---|---|---|
| **TrafficStream** | 14.22±0.13 | 23.16±0.27 | ✗ | ✗ |
| **STKEC** | 14.14±0.04 | 23.04±0.07 | ✗ | ✗ |
| **PECMP** | 14.85 * | 24.62 * | ✗ | ✗ |
| **TFMoE** | 14.18 * | 23.54 * | ✗ | ✗ |
| `EAC` | **13.53±0.06** | **21.77±0.10** | ✓ | ✓ |

**Few-Shot Performance.** While the current experiments encompass spatio-temporal datasets of various scales, we aim to examine more general and complex few-shot scenarios. Specifically, for continuous periodic spatio-temporal data, only a limited amount of training data is typically available for each period. We train or fine-tune the model using only 20% of the data for each period. As shown in Figure 5, we present the 12 step and average RMSE performance metrics across different years of the *PEMS-Stream* dataset. Our observations are as follows:

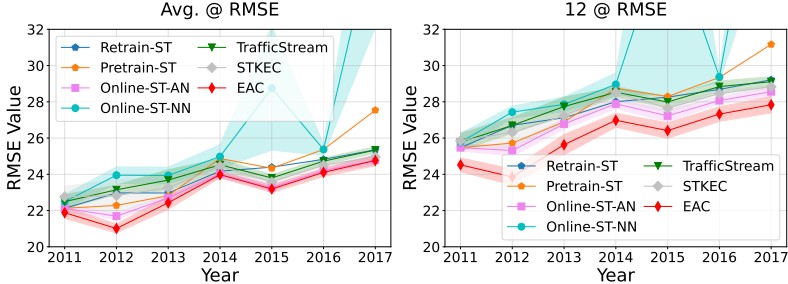

Figure 5: Few-Shot Scenario Forecasting in *PEMS-Stream* benchmark.

❶ **(Robustness)** All methods exhibit a decline in performance compared to the complete data scenario, with *Online-ST-NN* demonstrating significant sensitivity due to its inherent catastrophic forgetting issue. However, our `EAC` method exhibits controllable robustness across all years, outperforming all other methods. ❷ **(Adaptability)** The performance of all methods consistently declines as the periods extend, yet our `EAC` method demonstrates a relatively mild decline, particularly evident in the 12-step metrics, where it significantly outperforms all other approaches.

## 5.3 UNIVERSALITY STUDY (RQ2)

**Setting.** We further aim to demonstrate the universality of our EAC in enhancing the performance of various STGNN backbones, an aspect largely overlooked in previous studies. Specifically, STGNN can be categorized into spectral-based and spatial-based graph convolution operators, as well as recurrent-based, convolution-based, and attention-based sequence modeling operators. We select a representative operator from each category to form six distinct models, where the core architecture consists of two interleaved graph convolution modules and one sequence module. A detailed description of the different operators can be found in Appendix E. Additionally, we adapt different models to the prompt parameter pool proposed by EAC to compare the performance impact.

Table 3: Effect of EAC on the average performance of different STGNN component on the **PEMS-Stream**. C-based: Convolution-based, R-based: Recurrent-based, A-based: Attention-based.

| Methods | Spatial-based | | | | | | Spectral-based | | | | | |
| | C-based | | R-based | | A-based | | C-based | | R-based | | A-based | |
| Metric | MAE | RMSE | MAE | RMSE | MAE | RMSE | MAE | RMSE | MAE | RMSE | MAE | RMSE |
| w/o | 14.07±0.07 | 22.93±0.08 | 13.23±0.10 | 21.36±0.18 | 14.69±0.42 | 23.33±0.59 | 14.01±0.01 | 22.76±0.03 | 13.73±0.91 | 21.87±1.24 | 14.64±0.06 | 23.12±0.06 |
| EAC | 13.72±0.01 | 22.14±0.02 | **12.83**±0.05 | **20.59**±0.09 | 14.64±0.53 | 22.79±0.59 | 13.62±0.12 | 21.80±0.18 | 13.09±0.37 | 20.80±0.60 | 13.69±0.08 | 21.65±0.10 |
| Δ | - 2.48% | - 3.44% | - 3.02% | - 3.60% | - 0.34% | - 2.31% | - 2.78% | - 4.21% | - 4.66% | - 4.89% | - 6.48% | - 6.35% |

**Result Analysis.** Due to page limitations, we present MAE and RMSE metrics at average time steps using the *PEMS-Stream* dataset. As shown in Table 3, we observe that:

❶ Our EAC consistently demonstrates performance improvements across different combinations, highlighting its universality for various architectures. ❷ Compared to spatial domain-based graph convolution operators, our EAC shows a more pronounced enhancement for spectral domain-based methods. ❸ The advantages of recurrent-based sequence modeling operators are particularly evident, achieving the best results, while attention-based methods perform the worst. This aligns with intuition, as directly introducing vanilla multi-head attention may lead to excessive parameters and over-fitting; however, our approach still provides certain gains in these cases.

## 5.4 EFFICIENCY & LIGHTWEIGHT STUDY (RQ3 & RQ4)

**Overall Analysis.** We first conduct a comprehensive comparison of the EAC with other baselines in terms of performance, training speed, and memory usage. All models are configured with the same batch size to ensure fairness. As illustrated in Figure 6, we visualize the performance, average tuning parameters, and average training time (per period) of different methods on both the smallest dataset (*Energy-Stream*) and the largest dataset (*Air-Stream*). Our observations are as follows:

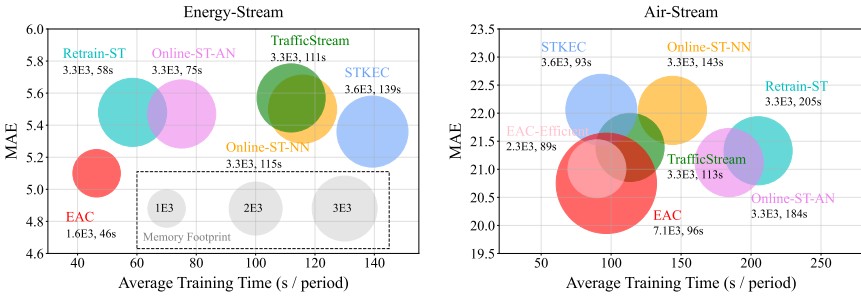

Figure 6: Efficiency & Lightwight & Performance Study.

❶ On datasets with a smaller number of nodes, our EAC consistently outperforms the others, achieving superior performance with only half the number of tuning parameters. Furthermore, the average training time per period accelerates by a factor of 1.26 to 3.02. In contrast, other methods such as *TrafficStream* and *STKEC*, despite employing second-order subgraph sampling techniques, do not benefit from efficiency improvements due to the limited number of nodes, which typically necessitates covering the entire graph. ❷ On datasets with a larger number of nodes, although the EAC exhibits a slightly higher number of tuning parameters, the freezing of the backbone model

results in faster training speeds while still achieving superior performance. ❸ According to the tuning principle of compression, we set $k = 2$ to replace 6, resulting in the `EAC` -*Efficient* version, which maintains relative performance superiority on larger datasets using only $\sim 63\%$ parameters compared to others. This demonstrates the superiority of the compression principle we propose.

**Hyper-parameter Analysis.** We further examine our sole hyper-parameter $k$, proposed through the compress principle, and its mutual influence on performance and tuning parameters. As shown in Figure 7, the horizontal axis denotes the values of $k$, while the vertical axes depict the performance on the *PEMS-Stream*. The color of the bars indicates the averaged percentage of tuning parameters relative to the total number of parameters across all periods. Our observations are as follows:

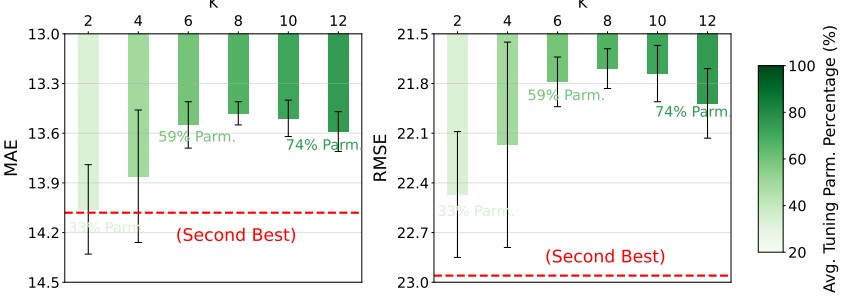

Figure 7: Hyper-parameter study in *PEMS-Stream* benchmark.

❶ From right to left, as the value of $k$ decreases, indicating a reduction in tuning parameters, the overall performance of the model deteriorates significantly, accompanied by increased volatility (*i.e.*, higher standard deviation). This aligns with our findings in Figure 4, as the effective representational information of the approximate prompt parameter pool is constrained with decreasing $k$, leading to a marked decline in performance. ❷ We set $k = 6$ as a default. Although performance continues to improve with increasing $k$, the gains are minimal. Moreover, excessively high values of $k$ may result in negative effects due to redundant parameters, leading to over-fitting. Consequently, our method achieves satisfactory performance using only approximately 59% of the tuning parameters, effectively balancing performance and parameter efficiency.

## 5.5 SIMPLICITY STUDY (RQ5)

**Simplicity Analysis.** Lastly, we aim to explore the simplicity of the node parameter prompt pool, as well as the effectiveness of the expansion and compression principle. We selected a common low-rank adaptation (*LoRA*) (Hu et al., 2021; Ruan et al., 2024) technique, which has

Table 4: Performance comparison of *LoRA-Based* method with `EAC` on *PEMS-Stream* benchmark.

| Model | 12 @ MAE | 12 @ RMSE | 12 @ MAPE | Avg. Time |
|---|---|---|---|---|
| *LoRA-Based* | 16.22±0.13 | 26.81±0.23 | 22.78±1.03 | 337.31±23.90 |
| EAC | **14.92**±0.11 | **24.17**±0.17 | **20.82**±0.16 | **224.33**±26.35 |

recently been widely used in large language models. Following the default architecture, we added low-rank adaptation layers to the sequence operators, setting the rank to 6, and fine-tuned the backbone model during each period. As shown in Table 4, we observe that simply applying *LoRA* layers without considering the specific spatio-temporal context of streaming parameters may not be highly effective. Moreover, our method enjoy shorter training times compared to *LoRA-based* approaches, further validating the superiority of our proposed expansion and compression tuning principle.

## 6 CONCLUSION

In this paper, we derive two fundamental tuning principle: expand and compress for continual spatio-temporal forecasting scenarios through empirical observation and theoretical analysis. Adhering to these principle, we propose a novel prompt-based continual forecasting method, `EAC` , which effectively adapts to the complexities of dynamic continual spatio-temporal forecasting problems. Experimental results across various datasets from different domains in the real world demonstrate that `EAC` possesses desirable characteristics such as simplicity, effectiveness, efficiency, and university. In the future, we will further explore large-scale pre-training methods for spatio-temporal data, leveraging the tuning principle proposed in this paper, which could broadly benefit a wide range of downstream continual forecasting tasks.

ACKNOWLEDGMENTS

The authors would like to thank the anonymous reviewers for their valuable comments. This work is mainly supported by the National Natural Science Foundation of China (No. 62402414). This work is also supported by the Guangzhou-HKUST(GZ) Joint Funding Program (No. 2024A03J0620), Guangzhou Municipal Science and Technology Project (No. 2023A03J0011), the Guangzhou Industrial Information and Intelligent Key Laboratory Project (No. 2024A03J0628), and a grant from State Key Laboratory of Resources and Environmental Information System, and Guangdong Provincial Key Lab of Integrated Communication, Sensing and Computation for Ubiquitous Internet of Things (No. 2023B1212010007).

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

# SUPPLEMENTARY MATERIAL
## EXPAND AND COMPRESS: EXPLORING TUNING PRINCIPLES FOR CONTINUAL SPATIO-TEMPORAL GRAPH FORECASTING

## TABLE OF CONTENTS

# A THEORETICAL PROOF

## A.1 PROMPT PARAMETER POOL CAN CONTINUOUSLY ADAPT TO HETEROGENEITY PROPERTY

*Proof.* First, based on the definition of the average node vector dispersion, the original feature matrix can be further rewritten as:

$$D(X) = \frac{1}{n^2} \sum_{i=1}^{n} \sum_{j=1}^{n} \|x_i - x_j\|^2. \tag{3}$$

For the spatio-temporal learning function $f_\theta$ with invariance (*i.e.*, a frozen STGNN backbone network), we have $X^\theta = f(\theta; X, P) = X + P^\theta$, if the network converges, $P^\theta$ can be obtained through nonlinear fitting. Therefore, for vectors in the matrix, we denote:

$$x^\theta = x + p^\theta. \tag{4}$$

Similarly, substituting this into the new feature matrix, it can be rewritten as:

$$D(X^\theta) = \frac{1}{n^2} \sum_{i=1}^{n} \sum_{j=1}^{n} \|x_i + p_i^\theta - x_j - p_j^\theta\|^2 \tag{5}$$

Based on the following equation property:

$$\sum_{i,j} \|x_i - x_j\|^2 = 2n \sum_{i=1}^{n} \|x_i\|^2 - 2 \left\| \sum_{i=1}^{n} x_i \right\|^2. \tag{6}$$

Thus, the difference between Equation 5 and Equation 3 can be rewritten as:

$$\begin{aligned}
&D(X^\theta) - D(X) \\
&= \frac{1}{n^2} \sum_{i=1}^{n} \sum_{j=1}^{n} \|x_i + p_i^\theta - x_j - p_j^\theta\|^2 - \frac{1}{n^2} \sum_{i=1}^{n} \sum_{j=1}^{n} \|x_i - x_j\|^2 \\
&= \frac{1}{n^2} \left( 2n \sum_{i=1}^{n} \|x_i + p_i^\theta\|^2 - 2 \left\| \sum_{i=1}^{n} (x_i + p_i^\theta) \right\|^2 - \left( 2n \sum_{i=1}^{n} \|x_i\|^2 - 2 \left\| \sum_{i=1}^{n} x_i \right\|^2 \right) \right) \\
&= \frac{2}{n} \sum_{i=1}^{n} \|x_i + p_i^\theta\|^2 - \frac{2}{n^2} \left\| \sum_{i=1}^{n} (x_i + p_i^\theta) \right\|^2 - \frac{2}{n} \sum_{i=1}^{n} \|x_i\|^2 + \frac{2}{n^2} \left\| \sum_{i=1}^{n} x_i \right\|^2.
\end{aligned} \tag{7}$$

The mean vectors of the original feature matrix and the prompt parameter matrix are defined as follows:

$$\mu = \frac{1}{n} \sum_{i=1}^{n} x_i, \quad \mu_p^\theta = \frac{1}{n} \sum_{i=1}^{n} p_i^\theta. \tag{8}$$

Therefore, we have:

$$\sum_{i=1}^{n} x_i = n\mu, \quad \sum_{i=1}^{n} p_i^\theta = n\mu_p^\theta. \tag{9}$$

Substituting this into Equation 7 and further simplifying yields:

$$
\begin{aligned}
D(X^\theta) &- D(X) \\
&= \frac{2}{n} \sum_{i=1}^n \|x_i + p_i^\theta\|^2 - \frac{2}{n^2} \|n\mu + n\mu_p^\theta\|^2 - \frac{2}{n} \sum_{i=1}^n \|x_i\|^2 + \frac{2}{n^2} \|n\mu\|^2 \\
&= \frac{2}{n} \sum_{i=1}^n \|x_i + p_i^\theta\|^2 - 2\|\mu + \mu_p^\theta\|^2 - \frac{2}{n} \sum_{i=1}^n \|x_i\|^2 + 2\|\mu\|^2 \\
&= \frac{2}{n} \left( \sum_{i=1}^n \|x_i\|^2 + \sum_{i=1}^n \|p_i^\theta\|^2 + 2\sum_{i=1}^n x_i^\top p_i^\theta \right) - 2 \left( \|\mu\|^2 + 2\mu^\top \mu_p^\theta + \|\mu_p^\theta\|^2 \right) - \frac{2}{n} \sum_{i=1}^n \|x_i\|^2 + 2\|\mu\|^2 \\
&= \frac{2}{n} \sum_{i=1}^N \|p_i^\theta\|^2 + \frac{4}{n} \sum_{i=1}^n x_i^\top p_i^\theta - 4\mu^\top \mu_p^\theta - 2\|\mu_p^\theta\|^2 \\
&= \frac{2}{n} \sum_{i=1}^n \|p_i^\theta\|^2 + \frac{4}{n} \left( \sum_{i=1}^n x_i \right)^\top \left( \frac{1}{n} \sum_{i=1}^n p_i^\theta \right) - 4\mu^\top \mu_p^\theta - 2\|\mu_p^\theta\|^2 \\
&= \frac{2}{n} \sum_{i=1}^n \|p_i^\theta\|^2 + \frac{4}{n} (n\mu)^\top \mu_p^\theta - 4\mu^\top \mu_p^\theta - 2\|\mu_p^\theta\|^2 \\
&= 2\left(\frac{1}{n} \sum_{i=1}^n \|p_i^\theta\|^2 - \|\mu_p^\theta\|^2\right).
\end{aligned}
\tag{10}
$$

According to the corollary of the Cauchy-Schwarz inequality (Horn & Johnson, 2012):

$$
\sum_{i=1}^n \|\mathbf{x}_i\|^2 \geq \frac{1}{n} \left\| \sum_{i=1}^n \mathbf{x}_i \right\|^2.
\tag{11}
$$

Therefore, for any set of vectors $\{p_i^\theta\}$, we have:

$$
\sum_{i=1}^n \|p_i^\theta\|^2 \geq \frac{1}{n} \left\| \sum_{i=1}^n p_i^\theta \right\|^2 = n\|\mu_p^\theta\|^2.
\tag{12}
$$

That is:

$$
\frac{1}{n} \sum_{i=1}^n \|p_i^\theta\|^2 - \|\mu_p^\theta\|^2 \geq 0.
\tag{13}
$$

Thus,

$$
D(X^\theta) - D(X) = 2\left(\frac{1}{n} \sum_{i=1}^n \|p_i^\theta\|^2 - \|\mu_p^\theta\|^2\right) \geq 0.
\tag{14}
$$

This completes the proof. $\qquad\square$

### A.2 Prompt Parameter Pool Can Continuously Satisfy the Low-rank Property

*Proof.* We first construct a Random Matrix $\Phi \in \mathbb{R}^{k \times n}$. Let $\Phi$ be a random matrix with entries $\phi_{ij}$ drawn independently from the standard normal distribution scaled by $1/\sqrt{k}$:

$$
\phi_{ij} \sim \mathcal{N}\left(0, \frac{1}{k}\right).
$$

Next, we can define matrices $A = \Phi^\top \in \mathbb{R}^{n \times k}$ and $B = \Phi P \in \mathbb{R}^{k \times d}$. Thus, we have

$$
AB = \Phi^\top (\Phi P) = (\Phi^\top \Phi) P.
$$

Consider the approximation error:

$$
\|P - AB\|_F = \left\|(I_n - \Phi^\top \Phi)P\right\|_F \leq \left\|I_n - \Phi^\top \Phi\right\|_2 \|P\|_F.
$$

Thus, our goal is transform to bound $\left\|I_n - \Phi^\top \Phi\right\|_2$. According to concentration inequality for tail bounds of spectral norms of sums of random matrices (Tropp, 2012), for any $\epsilon \in (0, 1)$:

$$\Pr\left(\left\|\Phi^\top \Phi - I_n\right\|_2 \geq \epsilon\right) \leq 2n \cdot e^{-\frac{k\epsilon^2}{2}}.$$

We further show that, when set $k = \left\lceil \frac{4 \log \min(n,d)}{\epsilon^2} \right\rceil$, we obtain:

$$\Pr\left(\left\|\Phi^\top \Phi - I_n\right\|_2 \geq \epsilon\right) \leq 2n \cdot e^{-2 \log \min(n,d)} = 2n \cdot \min(n, d)^{-2}.$$

Since $\min(n, d) \leq n$, it follows that:

$$2n \cdot \min(n, d)^{-2} \leq 2n \cdot n^{-2} = \frac{2}{n} \to 0 \quad \text{as} \quad n \to \infty.$$

We have:

$$\Pr\left(\left\|\Phi^\top \Phi - I_n\right\|_2 \leq \epsilon\right) \geq 1 - o(1).$$

Therefore, due to the symmetry of the matrix norm, with probability at least $1 - o(1)$, we have:

$$\|P - AB\|_F \leq \left\|I_n - \Phi^\top \Phi\right\|_2 \|P\|_F \leq \epsilon \|P\|_F.$$

Thus, there exist matrices $A \in \mathbb{R}^{n \times k}$ and $B \in \mathbb{R}^{k \times d}$ with $k = \mathcal{O}\left(\log(\min(n, d))\right)$ such that:

$$\Pr\left(\|P - AB\|_F \leq \epsilon \|P\|_F\right) \geq 1 - o(1),$$

where $o(1)$ represents a term that becomes negligible as $n$ grows large.

This completes the proof. □

## B  METHOD DETAIL

### B.1  METHOD ALGORITHM

---

**Algorithm 1** The workflow of `EAC` for continual spatio-temporal graph forecasting

---

**Input:**
    Dynamic streaming spatio-temporal graph $\mathbb{G} = (\mathcal{G}_1, \mathcal{G}_2, \cdots, \mathcal{G}_\mathcal{T})$
    Observation data $\mathbb{X} = (X_1, X_2, \cdots, X_\mathcal{T})$.
**Output:**
    A prompt parameter pool $\mathcal{P}$ (in memory).
**Pipeline:**
1: **while** Stream Graph $\mathbb{G}$ remains **do**
2:    **if** $\tau == 1$ **then**
3:       Construct an initial prompt parameter pool: $\mathcal{P} = A^{(\tau)}B$.
                                                                           ▷ *Tuning Principle II: Compress*
4:       Fusion of observed data $X$ and prompt parameter pool $\mathcal{P}$: $X_\tau = X_\tau + \mathcal{P}$.
5:       Jointly optimize the base STGNN $f_\theta$ and $\mathcal{P}$: $f_{\theta*} = \arg\min_\theta f_\theta(X_\tau, \mathcal{G}_1)$.
6:    **else**
7:       Reload the prompt pool $\mathcal{P}$ and model $f_{\theta*}$.
8:       Detect new nodes and construct a prompt parameter matrix $A^{(\tau)}$
9:       Add new node prompts to the prompt parameter pool: $\mathcal{P} = \mathcal{P}.append(A^{(\tau)}B)$
                                                                             ▷ *Tuning Principle I: Expand*
10:      Fusion of observed data $X$ and prompt parameter pool $\mathcal{P}$: $X_\tau = X_\tau + \mathcal{P}$.
11:      Jointly optimize the frozen STGNN $f_{\theta*}$ and $\mathcal{P}$ with data $X_\tau$.
12:    **end if**
13:    Use the STGNN and prompt parameter pool $\mathcal{P}$ for current period prediction.
14: **end while**
15: **return** Prompt Parameter Pool $\mathcal{P}$

---

Here, we present the workflow of `EAC` for continual spatio-temporal forecasting in Algorithm 1.

### B.2 METHOD PROCESS

Our input consists of a series of continual spatio-temporal graphs and corresponding observational data. Our objective is to maintain a continuous dynamic prompt parameter pool. Specifically, we first construct an initial prompt parameter pool $\mathcal{P}$ that includes the initial node prompt parameters $P^{(1)}$, which we approximate using subspace parameters $A^{(1)}$ and fixed parameters $B$ *(line 3)*. We then perform element-wise addition of the observational data $X_1$ with the prompt parameter pool at the node level *(line 4)*, and feed this into the base STGNN model $f_\theta$ for joint training *(line 5)*. In the subsequent $\tau$-th period, we detect new nodes compared to the $\tau - 1$ phase and construct a prompt parameter matrix $A^{(\tau-1)}$ for all newly added nodes *(line 8)*, which is then incorporated into the prompt parameter pool $\mathcal{P}$ *(line 9)*. We again perform element-wise addition of the observational data $D_\tau$ with the prompt parameter pool at the node level *(line 10)* and input this into the frozen base STGNN model $f_{\theta^*}$ for training *(line 11)*.

After completing each period of training, we use the current prompt parameter pool $P$ and the base STGNN $f_\theta$ to predict the results for the current period *(line 13)*.

## C EXPERIMENTAL DETAILS

### C.1 DATASETS DETAILS

Our experiments are carried out on three real-world datasets from diffrent domain. The statistics of these stream spatio-temporal graph datasets are shown in Table 5.

Table 5: Summary of datasets used for continual spatio-temporal datasets.

| Dataset | Domain | Time Range | Period | Node Evolute | Frequency | Frames |
|---------|--------|------------|--------|--------------|-----------|--------|
| *Air-Stream* | Weather | 01/01/2016 - 12/31/2019 | 4 | $1087 \rightarrow 1154$ $\rightarrow 1193 \rightarrow 1202$ | 1 hour | 34,065 |
| *PEMS-Stream* | Traffic | 07/10/2011 - 09/08/2017 | 7 | $655 \rightarrow 715 \rightarrow 786$ $\rightarrow 822 \rightarrow 834 \rightarrow 850$ $\rightarrow 871$ | 5 min | 61,992 |
| *Energy-Stream* | Energy | Unknown (245 days) | 4 | $103 \rightarrow 113$ $\rightarrow 122 \rightarrow 134$ | 10 min | 34,560 |

Specifically, we first utilize the *PEMS-Stream* dataset, which serves as a benchmark dataset, for our analysis. The objective is to predict future traffic flow based on historical traffic flow observations from a directed sensor graph. Additionally, we construct a *Air-Stream* dataset for the meteorological domain focused on air quality, aiming to predict future air quality index (AQI) flow based on observations from various environmental monitoring stations located in China. We segment the data into four periods, corresponding to four years. We also construct a *Energy-Stream* dataset for wind power in the energy domain, where the goal is to predict future indicators based on the generation metrics of a wind farm operated by a specific company (using the temperature inside the turbine nacelle as a substitute for active power flow observations). We also divided the data into four periods, corresponding to four years, based on the appropriate sub-period dataset size.

We follow conventional practices (Li et al., 2017; Bai et al., 2020) to define the graph topology for the air quality and wind power datasets. Specifically, we construct the adjacency matrix $A_\tau$ for $\tau$-th year using a threshold Gaussian kernel, defined as follows:

$$A_{\tau[ij]} = \begin{cases} \exp\left(-\frac{d_{ij}^2}{\sigma^2}\right) & \text{if } \exp\left(-\frac{d_{ij}^2}{\sigma^2}\right) \geq r \text{ and } i \neq j \\ 0 & \text{otherwise} \end{cases}$$

where $d_{ij}$ represents the distance between sensors $i$ and $j$, $\sigma$ is the standard deviation of all distances, and $r$ is the threshold. Empirically, we select $r$ values of 0.5 and 0.99 for the air quality and wind power datasets, respectively.

**Discussion:** There are some differences in the datasets of these three fields. Specifically, the primary difference between the three spatio-temporal datasets lies in the underlying spatio-temporal dynamics, or spatial dependencies: ❶ **(Differences in Underlying Spatio-temporal Dynamics)** In

order, *Energy-Stream* involves a small wind farm with closely spaced turbines that share similar spatial patterns. *PEMS-Stream* represents the entire transportation system in Southern California, with moderate spatial dependencies. *Air-Stream* goes further, encompassing air quality records from air monitoring stations across all of China, leading to much more complex spatial dependencies. These differences are reflected in the MAE and RMSE metrics, with the performance progressively degrading as the complexity increases. ❷ **(Special Characteristics of Energy Data)** Regarding the MAPE metric, we must note that in the *Energy-Stream* dataset, turbines occasionally face overheating protection or shutdown for inspection, which can cause some devices to produce very small values during certain time periods. This causes large percentage errors when using MAPE.

## C.2 BASELINE DETAILS

We have provided a detailed overview of various types of methods for continual spatio-temporal forecasting in the main body of the paper. The current core improvements primarily focus on online-based methods. Below is a brief introduction to these advance methods:

- TrafficStream (Chen et al., 2021b): The paper first introduces a novel forecasting framework leveraging spatio-temporal graph neural networks and continual learning to predict traffic flow in expanding and evolving networks. It addresses challenges in long-term traffic flow prediction by integrating emerging traffic patterns and consolidating historical knowledge through strategies like historical data replay and parameter smoothing. https://github.com/AprLie/TrafficStream/tree/main

- STKEC (Wang et al., 2023a): The paper also presents a continual learning framework, which is designed for traffic flow prediction on expanding traffic networks without the need for historical graph data. The framework introduces a pattern bank for storing representative network patterns and employs a pattern expansion mechanism to incorporate new patterns from evolving networks. https://github.com/wangbinwu13116175205/STKEC

- PECPM (Wang et al., 2023b): The paper also discusses a method for predicting traffic flow on graphs that change over time. To address the challenges presented by these dynamic graph, the paper proposes a framework that includes two main components: Knowledge Expansion and Knowledge Consolidation. The former is responsible for detecting and incorporating new patterns and structures as the graph expands. The latter aims to prevent the loss of knowledge about traffic patterns learned from historical data as the model updates itself with new information.

- TFMoE (Lee & Park, 2024): The paper introduces Traffic Forecasting Mixture of Experts for continual traffic forecasting on evolving networks. The key innovation is segmenting traffic flow into multiple groups and assigning a dedicated expert model to each group. This addresses the challenge of catastrophic forgetting and allows each expert to adapt to specific traffic patterns without interference.

| Horizon | 3 | | | 12 | | |
|---------|-----|------|---------|-----|------|---------|
| Metric | MAE | RMSE | MAPE(%) | MAE | RMSE | MAPE(%) |
| *ST-CRL* | 18.41 | 24.63 | 22.64 | 24.45 | 35.11 | 29.40 |
| *EAC* | 12.65±0.03 | 20.24±0.06 | 17.80±0.08 | 14.92±0.11 | 24.17±0.17 | 20.82±0.16 |

Table 6: Comparison of ST-CRL and our method on the PEMS-Stream benchmark.

We also highlight some related works and explain why they are not included as baselines:

- ST-CRL (Xiao et al., 2022): This work integrates reinforcement learning into continual spatio-temporal graph learning. We exclude it because its code is unavailable and its methods are not reproducible. Additionally, its poor performance further justifies this decision. Notably, according to the original paper, the comparison between ST-CRL and our method on the PEMS-Stream benchmark, as shown in Table 6, provides evidence for this.

- URCL (Miao et al., 2024): This work incorporates data augmentation into continual spatio-temporal graph learning. However, it essentially treats spatio-temporal graphs as static, with only observed instances evolving over time. Thus, this approach is not directly comparable to ours.

- UniST (Yuan et al., 2024): This work can be viewed as a parallel effort in two different directions. It focuses on large-scale pre-training in the initial stage. While it also uses some empirical prompt

learning methods for fine-tuning, it is not suitable for continuous incremental scenarios. Another key point is that it is limited to spatio-temporal grid data.

- FlashST (Li et al., 2024): This work is essentially limited to static spatio-temporal graphs, adjusting data distributions via prompt embedding regularization to achieve efficient model adaptation across different spatio-temporal prediction tasks. Therefore, it cannot be reasonably applied to continuous spatio-temporal graph learning.

- CMuST (Yi et al., 2024): This recent work addresses spatio-temporal learning in continuous multitask scenarios, enhancing individual tasks by jointly modeling learning tasks in the same spatio-temporal domain. However, this framework mainly focuses on task-level continual learning, transitioning from one task to another. It does not address the continuous spatio-temporal graph learning characteristics encountered in real spatio-temporal prediction scenarios. Hence, this approach is also not suitable for direct comparison to our single-task dynamic scenarios.

**Metrics Detail.** We use different metrics such as MAE, RMSE, and MAPE. Formally, these metrics are formulated as following:

$$\text{MAE} = \frac{1}{n}\sum_{i=1}^{n}|y_i - \hat{y}_i|, \quad \text{RMSE} = \sqrt{\frac{1}{n}\sum_{i=1}^{n}(y_i - \hat{y}_i)^2}, \quad \text{MAPE} = \frac{100\%}{n}\sum_{i=1}^{n}\left|\frac{\hat{y}_i - y_i}{y_i}\right|$$

where $n$ represents the indices of all observed samples, $y_i$ denotes the $i$-th actual sample, and $\hat{y}_i$ is the corresponding prediction.

Table 7: Hyperparameters setting.

| | |
|---|---|
| Graph operator hidden dimension | 64 |
| Sequence operator (TCN) kernel size | 3 |
| Graph layer | 2 |
| Sequence layer | 1 |
| Training epochs | 100 |
| Batch size | 128 |
| Adam $\epsilon$ | 1e-8 |
| Adam $\beta$ | (0.9,0.999) |
| Learning rate | 0.03 / 0.01 |
| Loss Function | MSE |
| Dropout | 0.1 / 0.0 |

**Parameter Detail.** Detailed hyperparameters settings are shown in Table 7. We use the same parameter configurations for our `EAC`, along with the other baseline methods according to the recommendation of previous studies Chen et al. (2021b); Wang et al. (2023a). Our only hyperparameter $k$ is set to 6 by default. All experiments are conducted on a Linux server equipped with a 1 × AMD EPYC 7763 128-Core Processor CPU (256GB memory) and 2 × NVIDIA RTX A6000 (48GB memory) GPUs. To carry out benchmark testing experiments, all baselines are set to run for a duration of 100 epochs by default, with specific timings contingent upon the method with early stop mechanism. *The source code, data, experimental results, logs, and model weights can be accessed in the anonymous repository at* `https://github.com/Onedean/EAC`.

## D   MORE RESULTS

To evaluate the performance differences between our `EAC` and models trained individually for each period, we compare the average forecasting performance over 12 time steps for each period on the *PEMS-Stream* dataset. This comparison includes our `EAC` and the individually trained *Retrain-ST* method, with results averaged over five random runs and shown in Table 8.

**Result Analysis: ❶** (Retrain-ST Underperformance): Retrain-ST exhibits unsatisfactory performance because it relies solely on limited data to train models for specific periods, failing to effectively leverage historical information from pre-trained models. In continuous spatiotemporal graph

| Methods | Metric | 2011 | 2012 | 2013 | 2014 | 2015 | 2016 | 2017 | Average |
|---|---|---|---|---|---|---|---|---|---|
| *Retrain-ST* | MAE | 14.26±0.13 | 13.69±0.26 | 13.88±0.18 | 14.76±0.11 | 14.14±0.19 | 13.70±0.15 | 15.26±0.57 | 14.24±0.12 |
| | RMSE | 21.97±0.21 | 21.60±0.42 | 22.50±0.27 | 23.82±0.19 | 23.15±0.25 | 24.40±0.26 | 24.98±0.62 | 23.20±0.16 |
| | MAPE(%) | 18.92±1.54 | 19.33±0.39 | 20.19±1.28 | 22.06±2.00 | 20.33±1.57 | 19.48±1.68 | 21.82±3.58 | 20.30±0.44 |
| `EAC` | MAE | 13.46±0.15 | 13.00±0.03 | 13.07±0.05 | 14.00±0.05 | 13.55±0.04 | 13.01±0.04 | 14.57±0.09 | 13.53±0.06 |
| | RMSE | 20.49±0.23 | 20.19±0.05 | 20.90±0.09 | 22.27±0.09 | 21.90±0.80 | 23.08±0.03 | 23.54±0.12 | 21.77±0.10 |
| | MAPE(%) | 17.85±0.35 | 18.12±0.42 | 18.51±0.17 | 20.04±0.40 | 19.30±0.31 | 17.86±0.17 | 21.16±0.29 | 18.98±0.08 |

Table 8: Comparison of the average performance of Retrain-ST and `EAC` across different periods in PEMS-Stream benchmark.

scenarios, underlying spatiotemporal dependencies are typically shared across periods. Leveraging historical training data helps models perform better, a benefit clearly observed with our EAC approach. ❷ (Performance Variability Across Different Periods): Performance differences across periods are evident even with the full dataset. These differences primarily stem from factors such as data noise and varying levels of learning difficulty in different periods.

**Further Analysis:** We visualize the noise levels across periods in the *PEMS-Stream* dataset, as shown in Figure 8. Specifically, Fourier transform is used to convert time series from the time domain to the frequency domain, where noise is typically represented by high-frequency components. A higher proportion of high-frequency energy indicates greater noise. Analyzing the frequency-domain characteristics of the spatio-temporal data reveals that for 2017 (red line), low-frequency components are minimal (Ratio > 0.42), while in 2014 (purple line), high-frequency components are more concentrated (0.46 ~ 0.48). This indicates that data from these periods are noisier and harder to learn, a trend consistent with the results in Table 8.

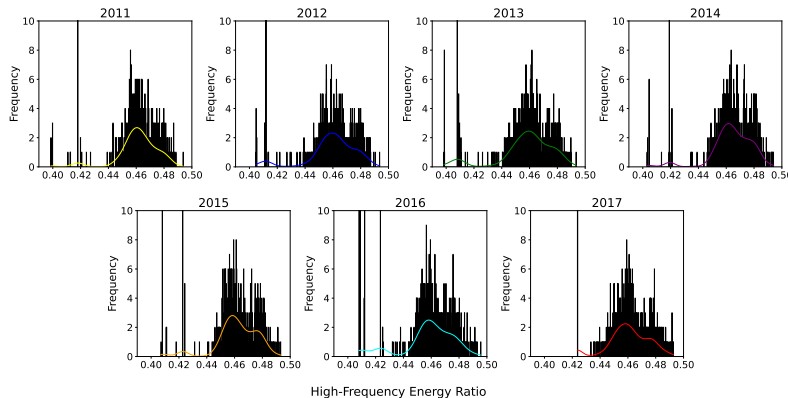

Figure 8: High-frequency energy ratios across different periods in PEMS-Stream benchmark.

# E STGNNS COMPONENT DETAILS

In the design of spatio-temporal graph neural network (STGNN), spectral-based and spatial-based graph convolutions have different operation modes, and sequence convolution operators can also be divided into recurrent-based, convolution-based, and attention-based. Below are the formulas and explanations of each type.

## E.1 SPECTRAL-BASED GRAPH CONVOLUTION FORMULA

Spectral graph convolution relies on the eigenvalue decomposition of the graph Laplacian matrix. The specific formula is as follows:

**Graph Convolution:**

$$\mathbf{Z} = \mathbf{U}g_\theta(\mathbf{\Lambda})\mathbf{U}^\top\mathbf{X}$$

- $\mathbf{U}$ is the matrix of eigenvectors of the graph Laplacian $\mathbf{L} = \mathbf{D} - \mathbf{A}$.

- $\mathbf{\Lambda}$ is the diagonal matrix of eigenvalues of the Laplacian.
- $\mathbf{X}$ is the node feature matrix.
- $g_\theta(\mathbf{\Lambda})$ is the convolution kernel function on the eigenvalues, typically parameterized (*e.g.*, using Chebyshev polynomial approximations).

**Approximation of Convolution Kernel:**

Using Chebyshev polynomial approximations, one can express the formula as:

$$\mathbf{Z} = \sum_{k=0}^{K} \theta_k \mathbf{T}_k(\tilde{\mathbf{L}})\mathbf{X}$$

- $\tilde{\mathbf{L}} = 2\mathbf{L}/\lambda_{\max} - \mathbf{I}$ is the normalized Laplacian matrix.
- $\mathbf{T}_k(\tilde{\mathbf{L}})$ is the Chebyshev polynomial, and $\theta_k$ are the learnable parameters.

### E.2 SPATIAL-BASED GRAPH CONVOLUTION FORMULA

Spatial graph convolution aggregates information directly using the adjacency matrix. The classic formula for spatial graph convolution is as follows:

$$\mathbf{Z} = \sigma\left(\hat{\mathbf{A}}\mathbf{X}\mathbf{W}\right)$$

- $\hat{\mathbf{A}} = \mathbf{D}^{-1/2}\mathbf{A}\mathbf{D}^{-1/2}$ is the normalized form of the adjacency matrix.
- $\mathbf{X}$ is the node feature matrix.
- $\mathbf{W}$ is the learnable weight matrix.
- $\sigma(\cdot)$ is a nonlinear activation function (*e.g.*, ReLU).

### E.3 RECURRENT-BASED SEQUENCE MODELING FORMULA

Recurrent-based sequence convolution typically uses recurrent neural networks (RNNs), such as LSTM or GRU. we choose LSTM, the formulas are as follows:

**LSTM Equations:**
- Forget Gate: $\mathbf{f}_t = \sigma\left(\mathbf{W}_f[\mathbf{h}_{t-1}, \mathbf{x}_t] + \mathbf{b}_f\right)$
- Input Gate: $\mathbf{i}_t = \sigma\left(\mathbf{W}_i[\mathbf{h}_{t-1}, \mathbf{x}_t] + \mathbf{b}_i\right)$
- Candidate Memory: $\tilde{\mathbf{C}}_t = \tanh\left(\mathbf{W}_C[\mathbf{h}_{t-1}, \mathbf{x}_t] + \mathbf{b}_C\right)$
- Output Gate: $\mathbf{o}_t = \sigma\left(\mathbf{W}_o[\mathbf{h}_{t-1}, \mathbf{x}_t] + \mathbf{b}_o\right)$
- Cell State Update: $\mathbf{C}_t = \mathbf{f}_t \odot \mathbf{C}_{t-1} + \mathbf{i}_t \odot \tilde{\mathbf{C}}_t$
- Hidden State Update: $\mathbf{h}_t = \mathbf{o}_t \odot \tanh(\mathbf{C}_t)$

### E.4 CONVOLUTION-BASED SEQUENCE MODELING FORMULA

Convolution-based sequence convolution employs one-dimensional (1D) convolution to process time series data. The formula is as follows:

$$\mathbf{Z} = \sigma\left(\text{Conv1D}(\mathbf{X}, \mathbf{W}, \mathbf{b})\right)$$

- $\mathbf{X} \in \mathbb{R}^{B \times C_{\text{in}} \times T}$ is the input sequence, where $B$ is the batch size, $C_{\text{in}}$ is the number of input channels, and $T$ is the time steps.
- $\mathbf{W} \in \mathbb{R}^{C_{\text{out}} \times C_{\text{in}} \times K}$ is the convolution kernel, with $K$ being the kernel size.
- $\mathbf{b}$ is the bias vector.
- $\sigma(\cdot)$ is the activation function.

### E.5 ATTENTION-BASED SEQUENCE MODELING FORMULA

The attention-based model commonly uses the formula for the multi-head self-attention mechanism (*e.g.*, in Transformers):

**Attention Weights:**

$$\text{Attention}(Q, K, V) = \text{softmax}\left(\frac{QK^\top}{\sqrt{d_k}}\right) V$$

- $Q = \mathbf{X}\mathbf{W}_Q$ is the query matrix.
- $K = \mathbf{X}\mathbf{W}_K$ is the key matrix.
- $V = \mathbf{X}\mathbf{W}_V$ is the value matrix.
- $d_k$ is the dimension of the key vectors.

**Multi-head Attention:**

$$\text{MultiHead}(Q, K, V) = \text{Concat}\left(\text{head}_1, \ldots, \text{head}_h\right) \mathbf{W}_O$$

Each head is defined as $\text{head}_i = \text{Attention}(Q_i, K_i, V_i)$.

### E.6 SUMMARY

- **Spectral-based Graph Convolution Operator**: Achieves graph convolution through the eigenvalue decomposition of the graph Laplacian.
- **Spatial-based Graph Convolution Operator**: Directly aggregates node features using the adjacency matrix.
- **Recurrent-based Sequence Operator**: Uses LSTM or GRU to capture temporal dependencies.
- **Convolution-based Sequence Operator**: Employs 1D convolution to handle time series data with fixed window sizes.
- **Attention-based Sequence Operator**: Uses multi-head self-attention to model global dependencies in long sequences.

These formulas form the core of spatio-temporal graph neural networks, allowing the combination of different operations to form the basis architecture of most current models.

## F MORE DISCUSSION

### F.1 DISCUSSION

**Parameter Inflation**: In addition to the analysis mentioned in the main text, we further provide a more detailed analysis of the concerns about parameter inflation. Specifically, this includes the following points:

- *(Adjustable Parameter Count)*: As we introduced in main text, there is no free lunch in this regard. While introducing the prompt parameter pool significantly improves performance, there is an inevitable risk of parameter inflation as the dataset size grows with more nodes. Therefore, we dedicated an entire section to empirical observations and theoretical analysis to explain how to reasonably eliminate redundant parameters, offering a compress principle. Consequently, the adjustable parameter count can be dynamically reduced with changes in $k$. In the hyperparameter analysis section, we further demonstrate that even with a limited number of tuning parameters, our approach still achieves SOTA performance. Thus, in large-scale scenarios, we can appropriately select the hyperparameter $k$ to balance performance and efficiency.

- *(Freezing Backbone Model)*: One of the main advantages of EAC is that, compared to existing methods, freezing the backbone model directly leads to significant efficiency improvements, even in large-scale datasets. For example, we maintained the fastest average training speed on the air-stream dataset, and this can be further accelerated by adjusting parameter $k$. Therefore, our approach is clearly the best choice compared to others.

- *(Advantages of In-Memory Storage)*: Another point worth noting is that we need to point out that our prompt parameter pool is separate from the backbone model. Therefore, we can naturally store it in memory and only load it when needed. Therefore, for practical applications, overloading the prompt parameter pool is unnecessary worry.

- *(Easy-to-manage solution)*: As for even larger global datasets, the current backbone model size would clearly be insufficient and would need to be scaled up. In contrast, the growth of the prompt parameter pool can be considered a more manageable solution.

**Use All Historical Data**: Stacking all historical data together for training might seem reasonable at first glance, but it is actually impractical and reflects a misunderstanding of continuous spatio-temporal graph learning. As we pointed out in the last sentence of the first paragraph of the introduction: *Due to computational and storage costs, it is often impractical to store all data and retrain the entire STGNN model from scratch for each time period.* Thus, the primary motivation behind continuous spatio-temporal graph modeling methods is:

- *(Training and Storage Costs)*: Storing all historical data and retraining is associated with unacceptable training and storage costs. Training costs are easy to understand, but storage costs are significant because the model is usually only a fraction of the size of the data (e.g., in the PEMS-Stream benchmark, the model size per year is 36KB compared to the dataset size of 1.3GB, approximately 1:37,865).

In addition to this fundamental motivation, we would like to share further insights:

- *(Privacy Risks)*: In common continuous modeling tasks such as vision and text, a key improvement direction is to avoid accessing historical data, as this poses privacy risks beyond storage costs (Wang et al., 2024d). Accessing models that store knowledge from historical data is clearly safer. Common improvements, such as regularization-based and prototype-based methods, are moving in this direction.

- *(Existing Approximation Methods)*: The existing *Online-ST* methods can be seen as an approximation solution to training with all historical data. However, this often suffers from catastrophic forgetting and the need for full parameter adjustments, issues that our `EAC` effectively addresses.

## F.2 LIMITATION

In this paper, we thoroughly investigate methods for continual spatio-temporal forecasing. Based on empirical observations and theoretical analysis, we propose two fundamental tuning principle for sustained training. In practice, we consider this kind of continual learning to fall under the paradigm of continual fine-tuning. However, given the superiority and generality of our approach, we believe this provides an avenue for future exploration of continual pre-training. While we have made a small step in this direction, several limitations still warrant attention.

❶ All current baselines and datasets primarily focus on scenarios involving the continuous expansion of spatio-temporal graphs, with little consideration given to their reduction. This focus is reasonable, as spatio-temporal data is typically collected from observation stations, which generally do not disappear once established. Consequently, existing streaming spatio-temporal datasets are predominantly expansion-oriented. Nonetheless, there are exceptional circumstances, such as monitoring anomalies at stations or the occurrence of natural disasters leading to station closures, resulting in node disappearance. But, we want to emphasize that our method can effectively handle such situations, as our prompt parameter pool design is node-level, allowing for flexible selection of target node sets for parameter integration.

❷ All current baselines and datasets span a maximum of seven years. We believe this is a reasonable constraint, as a longer time frame might lead to drastic changes in spatio-temporal patterns. A more practical approach would be to retrain the model directly in the current year. However, research extending beyond this time span remains worthy of exploration, which we leave for future work.

❸ Due to the node-level design of our prompt parameter pool, the issue of parameter inflation is inevitable. Although we have proposed effective compression principles in this paper, other avenues for compression, such as parameter sparsification and pruning, also merit investigation. Given that we have innovatively provided guiding principles for compression in this study, we reserve further improvement efforts for future research.

### F.3 FUTURE WORK

One lesson learned from our experiments is that the initially pre-trained spatio-temporal graph model is crucial for the subsequent continuous fine-tuning process. This is highly analogous to today's large language models, which compress rich intrinsic knowledge from vast corpora, allowing them to perform well with minimal fine-tuning in specific domains. Therefore, we consider a significant future research direction to be the training of a sufficiently large foundation spatio-temporal model from scratch, utilizing data from diverse fields and scenarios. While some discussions and studies have emerged recently, we believe that a truly foundation spatio-temporal model remains a considerable distance away. Thus, we view this as a long-term goal for future work.

