# OpenReview forum: "Expand and Compress: Exploring Tuning Principles for Continual Spatio-Temporal Graph Forecasting"
_ICLR.cc/2025/Conference — ICLR 2025 Poster_

### Official Review · Reviewer_YV6D · 2024-10-28

**Soundness:** 3
**Presentation:** 3
**Contribution:** 3
**Rating:** 8
**Confidence:** 5

**Summary:**

The paper addresses the challenges of continual learning in spatio-temporal forecasting, particularly for data streams that evolve due to the deployment of new sensors. Traditional spatio-temporal graph neural networks struggle with retraining inefficiencies and catastrophic forgetting when applied to such streaming data scenarios. To overcome these issues, the authors propose a novel method called EAC (Expand and Compress). The proposed approach enhances the model’s capacity to manage evolving data without the need for full retraining, ensuring efficient and effective handling of dynamic spatio-temporal data streams.

**Strengths:**

1. The paper addresses a highly practical problem in spatio-temporal graph forecasting. On the one hand, new sensors are deployed over time, on the other hand, the patterns of spatio-temporal dynamics evolve.

2. One of the strengths of the paper lies in its strong motivation, backed by both empirical and theoretical analysis.  The authors provide a clear rationale for addressing catastrophic forgetting and the challenges of handling dynamic, continuously evolving spatio-temporal data.

3. The proposed method is reasonable.  I appreciate the idea of fixing the backbone of the spatio-temporal graph model and updating the prompt pool.  This approach strikes a balance: on the one hand, it preserves knowledge from previously trained samples, and on the other hand, it adapts to new incoming data effectively.

**Weaknesses:**

1. The design of the prompt pool is not clearly explained. Specifically, it is unclear what the prompt pool contains and how exactly these prompts are utilized within the model. Additionally, there is a lack of clarity on how the system handles the incorporation of new sensors in dynamic environments, which is a crucial aspect of the proposed approach.

2. The evaluation lacks a comparison with models trained separately for each period. While the proposed continual learning method shows promising results, it is essential to establish a performance upper bound by comparing it to a scenario where separate models are trained for different periods.

3. While the method outperforms baselines, I am concerned about its long-term effectiveness. In Figure 5, the model’s performance shows significant degradation over time, with the RMSE increasing from 24 to 28—indicating a more than 10% reduction in performance. Although this is in the context of few-shot learning, I suspect a similar trend would be observed in non-few-shot scenarios as well.
While separate training for each period may be more time-consuming, it could potentially achieve better performance, and it is storage-efficient since only the latest model needs to be saved. Therefore, it is crucial to assess whether the trade-off between reduced performance and computational efficiency is truly justified.

**Questions:**

See weaknesses

---

> ### Author Response · Authors · 2024-11-15
> **The First Part of the Response to Reviewer YV6D**
>
> Dear Reviewer YV6D,
>
> We sincerely thank you for taking the time and effort to provide valuable feedback on our paper. We apologize for any misunderstandings caused. We have carefully considered each of your points and have addressed them one by one.
>
> ---
>
> > **Q1: The design of the prompt pool is not clearly explained, and there is a lack of clarity on how the system handles the integration of new sensors in a dynamic environment.**
> >
> - We are sorry to hear that. A gentle reminder: you may have _**missed the pseudocode and detailed description of our algorithmic workflow in Appendix B**_.
> - Furthermore, the way the system handles sensors in dynamic environments is precisely the role of the prompt parameters added to our prompt pool. As explained in Section 4.1, through detailed empirical observations and theoretical analysis, we derive principle 1, which shows that the prompt pool can continuously adapt to the heterogeneity of spatio-temporal data generated by new sensors in dynamic environments.
> - We sincerely hope these clarifications help clear up any misunderstandings.
>
> ---
>
> > **Q2: The evaluation lacks comparison with models trained separately for each period. It is necessary to establish performance upper bounds by comparing with the scenario where models are trained separately for each period.**
> >
> - We apologize for the confusion. A gentle reminder: you may have _**missed our description of the Retrain-ST method**_, which is precisely designed for the scenario where separate models are trained for each period. Regarding why its performance is not satisfactory, we have provided a detailed analysis in Section 5.2 of the paper. However, we will summarize and address it here for your quick understanding:
>     - The Retrain-ST method tends to show unsatisfactory results mainly because it relies solely on limited data to train period-specific models, without effectively utilizing the historical information from the pre-trained model. In continuous spatio-temporal graph scenarios, underlying spatio-temporal dependencies are shared, and more historical training data typically helps the model achieve better performance.
>
> - _**If your concern is related to the stacking of all historical data to train the model**_, we would also like to clarify that while stacking all historical data may sound reasonable at first, it is actually impractical and represents a misunderstanding of continuous spatio-temporal graph learning. As pointed out in the last sentence of the introduction: “Due to computational and storage costs, it is often impractical to store all data and retrain the entire STGNN model from scratch for each time period." Therefore, the motivation behind all continuous spatio-temporal graph modeling methods is as follows:
>
>   - **(Training and Storage Costs):** Storing all historical data and retraining is associated with unacceptable training and storage costs. Training costs are easy to understand, but storage costs are significant because the model is usually only a fraction of the size of the data (e.g., in the PEMS-Stream benchmark, the _**model size**_ per year is  _**36KB**_ compared to the _**dataset size**_ of _**1.3GB**_, approximately _**37,865:1**_).
>
>   - In addition to this fundamental motivation, we would like to share further insights:
>
>     - **(Privacy Risks):** In common continuous modeling tasks such as vision and text, a key improvement direction is to avoid accessing historical data, as this poses privacy risks beyond storage costs [1]. Accessing models that store knowledge from historical data is clearly safer. Common improvements, such as regularization-based and prototype-based methods, are moving in this direction.
>
>     - **(Practical Impossibility):** Unlike vision and text tasks, spatio-temporal graphs have a unique property: their nodes are constantly changing. This introduces practical issues that make it nearly impossible to implement. For example, when training a neural network, _**data must be fixed into a certain format for each batch**_ to fully leverage GPU batch processing capabilities. The number of nodes in spatio-temporal graphs changes across different periods, _**making this impractical**_. Therefore, most methods seek a backbone STGNN independent of node count to accept spatio-temporal graph data from different time periods, but it still requires that node counts must be consistent during training within the same period.
>     - **(Existing Approximation Methods):** The existing Online-ST methods can be seen as an approximation solution to training with all historical data. However, this often suffers from catastrophic forgetting and the need for full parameter adjustments, issues that our EAC effectively addresses.
>
> - We sincerely hope these insights help clear up the misunderstanding.
>
> [1] Wang, et al. "A comprehensive survey of continual learning: theory, method and application." *IEEE TPAMI,* 2024.

---

> ### Author Response · Authors · 2024-11-15
> **The Second Part of the Response to Reviewer YV6D**
>
> ---
>
> > **Q3: Long-term effectiveness: Will similar trends be observed in non-small-sample scenarios? Is training separately for each period likely to achieve better performance?**
> >
> - We apologize for any unnecessary misunderstandings. This issue likely arises from our attempt to be fair and compress enough comparative information in Table 1. Below, we provide a comparison between the Retrain-ST method (which trains separate models for each period) and our method on the PEMS-Stream dataset, comparing performance over avg. 12-step predictions across multiple periods (averaged over five random runs):
>
> | Methods | Metric | 2011 | 2012 | 2013 | 2014 | 2015 | 2016 | 2017 | Average |
> | --- | --- | --- | --- | --- | --- | --- | --- | --- | --- |
> | Retrain-ST | MAE | 14.26±0.13 | 13.69±0.26 | 13.88±0.18 | 14.76±0.11 | 14.14±0.19 | 13.70±0.15 | 15.26±0.57 | 14.24±0.12 |
> | Our | MAE | 13.46±0.15 | 13.00±0.03 | 13.07±0.05 | 14.00±0.05 | 13.55±0.04 | 13.01±0.04 | 14.57±0.09 | 13.53±0.06 |
> | Retrain-ST | RMSE | 21.97±0.21 | 21.60±0.42 | 22.50±0.27 | 23.82±0.19 | 23.15±0.25 | 24.40±0.26 | 24.98±0.62 | 23.20±0.16 |
> | Our | RMSE | 20.49±0.23 | 20.19±0.05 | 20.90±0.09 | 22.27±0.09 | 21.90±0.80 | 23.08±0.03 | 23.54±0.12 | 21.77±0.10 |
> | Retrain-ST | MAPE | 18.92±1.54 | 19.33±0.39 | 20.19±1.28 | 22.06±2.00 | 20.33±1.57 | 19.48±1.68 | 21.82±3.58 | 20.30±0.44 |
> | Our | MAPE | 17.85±0.35 | 18.12±0.42 | 18.51±0.17 | 20.04±0.40 | 19.30±0.31 | 17.86±0.17 | 21.16±0.29 | 18.98±0.08 |
> - For large datasets, performance varies across periods mainly due to noise and difficulty levels in the current period’s dataset. As for retraining separately, it actually doesn’t perform well, as explained in the previous response. Also, retraining typically requires starting from scratch. By inheriting previous weights, we can significantly accelerate the optimization process, as shown in the table below, where we compare the average training time per period between our method and retraining each period separately (averaged over five random runs on the PEMS-Stream dataset):
>
> | Method | Training Time (s) / Period |
> | --- | --- |
> | Retrain-ST | 511.44±27.69 |
> | Our | 224.33±26.35 |
>
> ---
>
> **We greatly appreciate your valuable feedback**, and we will incorporate these detailed discussions into the final revision of the paper. We hope the above answers help address your concerns. **If possible, we kindly request you to reconsider raising the score.** If you have any further suggestions, we would be more than happy to discuss them and make necessary improvements to the paper.
>
> Best regards,
>
> All authors

---

> ### Comment · Reviewer_YV6D · 2024-11-20
>
> I appreciate the authors' effort put into addressing my concerns.  Regarding the prompt pool design, while Appendix provide explanations, the clarity of the integration process for new sensors in dynamic environments could be improved. I recommend revising the main text to explicitly highlight this process, potentially with the inclusion of a simplified diagram to enhance understanding.
>
> I also suggest incorporating the results of separate period evaluations into the revised paper, perhaps in the appendix. Including a brief explanation of how noise and varying difficulty levels affect performance across periods, along with a discussion of why the proposed approach outperforms retraining for each period, would further strengthen the paper's contribution.
>
> If these revisions are addressed, I will raise my score.

---

> > ### Author Response · Authors · 2024-11-21
> >
> > Dear Reviewer YV6D,
> >
> > We sincerely appreciate your recognition of our hard work. We deeply value the time and effort you have invested in providing us with insightful feedback on our paper. **Following your suggestions, we have revised and uploaded the updated version.** **Below is a brief summary of the changes for you:**
> >
> > + **Prompt Pool Design**:
> >    + We revised Section 4 to explicitly emphasize the learning process of the prompt parameter pool. Additionally, we updated Figure 2 to enhance clarity and understanding.
> >
> > + **Evaluation of Individual Periods**:
> >    + Beyond our previous response, we further visualized the noise levels of each period in the PEMS-Stream dataset, as shown in Figure 8. Specifically, we applied Fourier transform to convert the time series from the time domain to the frequency domain, where noise is typically represented by high-frequency components. A higher proportion of high-frequency energy indicates greater noise. By analyzing the frequency-domain characteristics of the spatio-temporal data, we observed that in 2017 (red line), low-frequency components are minimal (Ratio > 0.42), while in 2014 (purple line), high-frequency components are more concentrated (0.46–0.48). This suggests that data from these periods exhibit higher noise levels, making them harder to learn and resulting in poorer average prediction performance. This trend aligns with the results presented in Table 8.
> >
> > We sincerely hope that these revisions resolve your issues, and thank you again!
> >
> > Best,
> >
> > Authors.

---

> > > ### Comment · Reviewer_YV6D · 2024-11-25
> > >
> > > I appreciate the authors' revision, and I have raised my score.

---

> > > > ### Author Response · Authors · 2024-11-25
> > > > **Sincere Gratitude from Authors**
> > > >
> > > > We are delighted that our responses have successfully addressed all of your concerns. We would like to express our deepest gratitude for taking the time to review our paper and for providing such detailed and invaluable feedback.
> > > >
> > > > Best wishes, Authors of submission 3085

---

### Official Review · Reviewer_vM4w · 2024-10-31

**Soundness:** 3
**Presentation:** 3
**Contribution:** 3
**Rating:** 8
**Confidence:** 4

**Summary:**

This paper introduces a novel framework, EAC, for continual spatio-temporal graph forecasting. The authors address the challenges of retraining inefficiency and catastrophic forgetting in streaming spatio-temporal data scenarios by proposing a prompt tuning-based approach. They present two tuning principles—expand and compress—that guide both empirical and theoretical analysis. The expand principle addresses the dynamic heterogeneity of the data, while the compress principle tackles parameter inflation. Results demonstrate that EAC is effective, efficient, universal, and lightweight in tuning, with extensive experiments on real-world datasets supporting these claims.

**Strengths:**

- The problem addressed is significant, as spatio-temporal graph forecasting has applications in areas such as traffic flow and air quality monitoring. The proposed solution offers potential improvements in efficiency and model effectiveness in dynamic, real-world environments compared to previous methods.

- The paper presents a novel approach to continual learning within the context of spatio-temporal graph forecasting. The exploration of prompt tuning principles is innovative, and the authors offer a detailed and well-supported discussion of existing paradigms along with an extensive experimental analysis.

- The methodology is well-developed, with clear explanations of the theoretical foundations and empirical insights leading to the expand and compress tuning principles. While node-level parameters and low-rank decomposition are common in the field, the authors’ thorough analysis and discussion bring valuable new perspectives.

- The paper is well-organized and clearly written, making complex concepts accessible. The figures and tables are clear and complement the textual explanations effectively.

**Weaknesses:**

- While the prompt-based tuning paradigm for continual spatio-temporal forecasting is novel, similar recent methods [1,2,3] are only briefly mentioned in related work. A more detailed discussion of these approaches and their connection to the present work would be beneficial.

[1] Yuan, Yuan, et al. "Unist: a prompt-empowered universal model for urban spatio-temporal prediction." SIGKDD, 2024.

[2] Li, Zhonghang, et al. "FlashST: A Simple and Universal Prompt-Tuning Framework for Traffic Prediction." ICML, 2024.

[3] Yi, Zhongchao, et al. "Get Rid of Task Isolation: A Continuous Multi-task Spatio-Temporal Learning Framework." NIPS, 2024.

- The approach still has several limitations, such as performance over long time spans and parameter inflation. However, the authors appropriately address these limitations in detail in the appendix.

**Questions:**

- The authors note that the choice of pre-training backbone model is crucial. Does this imply that their method is more effective with larger-scale STGNN backbones?

- How would the EAC model adapt if the graph were to shrink, for instance, due to the removal of sensors or monitoring stations? Why was this scenario not included in the comparisons?

---

> ### Author Response · Authors · 2024-11-15
> **The First Part  of the Response to Reviewer vM4w**
>
> Dear Reviewer vM4w,
>
> We sincerely thank you for your hard work on our paper. We carefully considered each of your comments and dealt with them accordingly.
>
> ---
>
> > **Q1: Discussing more related works?**
>
> - We are happy to provide further discussion on these works. Specifically:
>     - **Unist [1]**: This work can be viewed as a parallel effort in two different directions. It focuses on large-scale pretraining in the initial stage. While it also uses some empirical prompt learning methods for fine-tuning, _**it is not suitable for continuous incremental scenarios**_. Another key point is that it is limited to spatiotemporal grid data and cannot be applied to spatiotemporal graph scenarios.
>     - **FlashST [2]**: This work is essentially limited to static spatiotemporal graphs, adjusting data distributions via prompt embedding regularization to achieve efficient model adaptation across different spatiotemporal prediction tasks. _**Therefore, it cannot be reasonably applied to continuous spatiotemporal graph learning.**_
>     - **CMuST [3]**: This recent outstanding work addresses spatiotemporal learning in continuous multitask scenarios, enhancing individual tasks by jointly modeling learning tasks in the same spatiotemporal domain. However, this framework mainly focuses on task-level continual learning, transitioning from one task to another. It does not address the continuous spatiotemporal graph learning characteristics encountered in real spatiotemporal prediction scenarios. _**Hence, this approach is also not suitable for direct comparison to our single-task dynamic forecasting scenarios.**_
>     - We hope these insights help clarify the unique aspects of our work.
>
> [1] Yuan, Yuan, et al. "Unist: a prompt-empowered universal model for urban spatio-temporal prediction." SIGKDD, 2024.
>
> [2] Li, Zhonghang, et al. "FlashST: A Simple and Universal Prompt-Tuning Framework for Traffic Prediction." ICML, 2024.
>
> [3] Yi, Zhongchao, et al. "Get Rid of Task Isolation: A Continuous Multi-task Spatio-Temporal Learning Framework." NIPS, 2024.
>
> ---
>
> > **Q2: The issue of parameter explosion needs to be discussed.**
>
> - We are happy to clarify further. In addition to the analyses in the paper, here are deeper insights:
>   - **(Adjustable Parameter Count):** As we introduced in Section 4.2, there is no free lunch in this regard. We acknowledge that while introducing the prompt parameter pool significantly improves performance, there is an inevitable risk of parameter bloat as the dataset size grows with more nodes. Therefore, we dedicated an entire section to empirical observations and theoretical analysis to explain how to reasonably eliminate redundant parameters, offering a _**compress principle**_. Consequently, the adjustable parameter count can be dynamically reduced with changes in _**k**_. In the hyperparameter analysis section, we further demonstrate that even with a limited number of tuning parameters, our approach still achieves SOTA performance. Thus, in large-scale scenarios, we can appropriately select the hyperparameter _**k**_ to balance performance and efficiency.
>   - **(Freezing Backbone Model):** One of the main advantages of EAC is that, compared to existing methods, freezing the backbone model directly leads to significant efficiency improvements, even in large-scale datasets. For example, in Figure 6, we maintained the fastest average training speed on the *air-stream* dataset, and this can be further accelerated by adjusting parameter k. Therefore, our approach is clearly the best choice compared to others.
>   - **(Advantages of In-Memory Storage):** Another point worth noting is that we need to point out that our prompt parameter pool is separate from the backbone model. Therefore, we can naturally store it in memory and only load it when needed. Therefore, for practical applications, overloading the prompt parameter pool is unnecessary worry.
>   - **(New Largest-Scale Benchmark):** We also would like to point out that the current mainstream benchmark for continuous spatio-temporal learning is *PEMS-Stream*, which has over 800 nodes. In this paper, we further gathered and constructed benchmark datasets from various domains (including meteorology and energy) and different scales (with more and fewer nodes), aiming to provide a richer evaluation for future work. Notably, *Air-Stream* includes spatio-temporal data from air monitoring stations across China, which we believe is practical for deployment. As for even larger global datasets, the current backbone model size would clearly be insufficient and would need to be scaled up, as seen in current large-scale spatio-temporal models [4, 5]. In contrast, the growth of the prompt parameter pool can be considered a more manageable solution.
>
> [4] Lam, et al. "GraphCast: Learning skillful medium-range global weather forecasting." *Science,*, 2023.
>
> [5] Shi, et al. "Time-MoE: Billion-Scale Time Series Foundation Models with Mixture of Experts." *arXiv,* 2024.

---

> ### Author Response · Authors · 2024-11-15
> **The Second Part of the Response to Reviewer vM4w**
>
> ---
>
> > **Q3: Would EAC benefit from large-scale STGNN backbones?**
>
> Yes, although we don’t emphasize this, we do mention that the results in Table 1 are not our upper limit. For example, in Table 3, by changing the backbone, the performance of our method improves further. We believe this benefit is not captured by other methods.
>
> ---
>
> > **Q4: How does EAC adapt to graph reduction, and why was this scenario not compared?**
>
> We are happy to clarify to avoid any misunderstandings. As discussed in the appendix, our method easily adapts to graph reduction scenarios. EAC uses node-level prompts, so for nodes that disappear in a new spatiotemporal graph, we simply do not load the corresponding prompt parameters. The reason this scenario wasn't compared is twofold:
> - **(No Suitable Real-world Datasets):** Firstly, in real-world observation stations, once established, they are rarely removed, so there is almost no real-world spatio-temporal graph reduction dataset.
> - **(No Suitable Comparison Baselines):** Secondly, current baselines cannot handle this setup, so we omitted this comparison.
>
> ---
>
> Thank you for your valuable feedback. We will incorporate these detailed discussions into the final revised version. Once again, **we appreciate your guidance!**

---

> ### Author Response · Authors · 2024-11-21
> **Kindly Request for Reviewer's Feedback**
>
> Dear Reviewer vM4w,
>
> **Since the End of the Rebuttal is coming very soon - only a few days left, we would like to inquire if our response addresses your primary concerns.** If you have any additional suggestions, we are more than willing to engage in further discussions and make necessary improvements to the paper.
>
> Thanks again for dedicating your time to enhancing our paper!
>
> Looking forward to your feedback.
>
> Best,
>
> Authors

---

### Official Review · Reviewer_7vMi · 2024-11-01

**Soundness:** 4
**Presentation:** 4
**Contribution:** 4
**Rating:** 8
**Confidence:** 5

**Summary:**

This paper introduces a prompt-tuning approach for continual spatio-temporal graph forecasting, specifically addressing the challenges of dynamic data streams. The authors propose the EAC framework, guided by two tuning principles, "Expand" and "Compress," to handle continual learning in STGNNs. By utilizing a continual prompt pool, EAC allows the base STGNN to accommodate new data while minimizing catastrophic forgetting. The authors demonstrate the approach’s effectiveness across various datasets, showcasing improvements in efficiency and adaptability compared to other methods.

**Strengths:**

S1.  EAC’s application of prompt tuning principles in continual spatio-temporal forecasting is novel, integrating dynamic prompt pool adjustments to effectively handle incoming data.
S2. The methodology is backed by both empirical and theoretical analysis, and the explanations are clear.
S3. The experimental results are impressive.

**Weaknesses:**

W1: While EAC is compared with several traditional and just-in-time tuning baselines, it is not included in comparison with other recent continuous learning techniques, such as combinations with reinforcement learning (Xiao et al., 2022) and data augmentation (Miao et al., 2024) mentioned in RELATED WORK. The reasons for the missing baselines are required.
W2: The Prompt Parameter Pool in EAC may introduce an issue of parameter bloat, which needs to be discussed.

**Questions:**

See the above.

---

> ### Author Response · Authors · 2024-11-15
> **The Response to Reviewer 7vMi**
>
> Dear Reviewer 7vMi,
>
> We sincerely appreciate the time and effort you have spent providing insightful feedback on our paper. We are honored that you recognized our hard work. We have carefully considered each of your comments and have addressed them one by one.
>
> ---
>
> > **Q1: Discussing more related works, and why they were not compared?**
>
> - We would be happy to clarify this point. Specifically:
>   - Regarding the work combining reinforcement learning, ST-CRL [1], we did not include it because of its inaccessible code and non-reproducible methodological examples. Additionally, another consideration was its poor results. Notably, in the original paper, the comparison between ST-CRL and our method on the PEMS-Stream benchmark is as follows (Table 1):
>
>   | Horizon |   | 3  |   |  | 12  | |
>   | ---       | --- | --- | --- | --- | --- | --- |
>   |Metric    | MAE | RMSE | MAPE | MAE | RMSE | MAPE |
>   | ST-CRL    | 18.41 | 24.63 | 22.64 | 24.45 | 35.11 | 29.40 |
>   | Our       | 12.65±0.03 | 20.24±0.06 | 17.80±0.08 | 14.92±0.11 | 24.17±0.17 | 20.82±0.16 |
>
>   - Regarding the work combining data augmentation, URCL [2], this work essentially treats the spatio-temporal graph as static, with only the observed instances changing over time. Therefore, this method cannot be directly compared with ours.
>
> We hope these insights help clarify any misunderstandings.
>
> [1] Xiao, et al. "Streaming Traffic Flow Prediction Based on Continuous Reinforcement Learning," ICDMW, 2022.
>
> [2] Miao, et al. "A unified replay-based continuous learning framework for spatio-temporal prediction on streaming data." ICDE, 2024.
>
> ---
>
> > **Q2: The issue of parameter explosion needs to be discussed.**
>
> - We are happy to provide further clarification. Beyond the analysis in the paper, here are some deeper insights:
>
>   - **(Adjustable Parameter Count):** As we introduced in Section 4.2, there is no free lunch in this regard. We acknowledge that while introducing the prompt parameter pool significantly improves performance, there is an inevitable risk of parameter bloat as the dataset size grows with more nodes. Therefore, we dedicated an entire section to empirical observations and theoretical analysis to explain how to reasonably eliminate redundant parameters, offering a _**compress principle**_. Consequently, the adjustable parameter count can be dynamically reduced with changes in _**k**_. In the hyperparameter analysis section, we further demonstrate that even with a limited number of tuning parameters, our approach still achieves SOTA performance. Thus, in large-scale scenarios, we can appropriately select the hyperparameter _**k**_ to balance performance and efficiency.
>   - **(Freezing Backbone Model):** One of the main advantages of EAC is that, compared to existing methods, freezing the backbone model directly leads to significant efficiency improvements, even in large-scale datasets. For example, in Figure 6, we maintained the fastest average training speed on the *air-stream* dataset, and this can be further accelerated by adjusting parameter k. Therefore, our approach is clearly the best choice compared to others.
>   - **(Advantages of In-Memory Storage):** Another point worth noting is that we need to point out that our prompt parameter pool is separate from the backbone model. Therefore, we can naturally store it in memory and only load it when needed. Therefore, for practical applications, overloading the prompt parameter pool is unnecessary worry.
>   - **(New Largest-Scale Benchmark):** We also would like to point out that the current mainstream benchmark for continuous spatio-temporal learning is *PEMS-Stream*, which has over 800 nodes. In this paper, we further gathered and constructed benchmark datasets from various domains (including meteorology and energy) and different scales (with more and fewer nodes), aiming to provide a richer evaluation for future work. Notably, *Air-Stream* includes spatio-temporal data from air monitoring stations across China, which we believe is practical for deployment. As for even larger global datasets, the current backbone model size would clearly be insufficient and would need to be scaled up, as seen in current large-scale spatio-temporal models [3, 4]. In contrast, the growth of the prompt parameter pool can be considered a more manageable solution.
>
> [3] Lam, et al. "GraphCast: Learning skillful medium-range global weather forecasting." *Science,*, 2023.
>
> [4] Shi, et al. "Time-MoE: Billion-Scale Time Series Foundation Models with Mixture of Experts." *arXiv,* 2024.
>
>
> ---
>
> Thank you for your valuable feedback. We will appropriately incorporate these detailed discussions into the final revised version of our paper. Once again, **we appreciate your guidance**!

---

> > ### Comment · Reviewer_7vMi · 2024-11-26
> >
> > Thanks for the efforts! The authors have addressed all the concerns.

---

> > > ### Author Response · Authors · 2024-11-26
> > > **Thank you for your time and consideration**
> > >
> > > Dear Reviewer 7vMi,
> > >
> > > We are glad to hear that our rebuttal effectively addressed your concerns. Thank you again for taking the time and effort to provide valuable feedback on our paper.
> > >
> > > Best,
> > >
> > > Authors

---

> ### Author Response · Authors · 2024-11-21
> **Kindly Request for Reviewer's Feedback**
>
> Dear Reviewer 7vMi,
>
> **Since the End of the Rebuttal is coming very soon - only a few days left, we would like to inquire if our response addresses your primary concerns.** If you have any additional suggestions, we are more than willing to engage in further discussions and make necessary improvements to the paper.
>
> Thanks again for dedicating your time to enhancing our paper!
>
> Looking forward to your feedback.
>
> Best,
>
> Authors

---

### Official Review · Reviewer_S8oR · 2024-11-02

**Soundness:** 2
**Presentation:** 3
**Contribution:** 2
**Rating:** 3
**Confidence:** 5

**Summary:**

This paper proposes EAC, a continuous spatio-temporal graph forecasting framework based on a continuous prompt parameter pool, aiming to address prediction challenges in dynamic streaming spatio-temporal data. EAC’s core idea is to freeze the base STGNN model and dynamically adjust the prompt parameter pool to adapt to new node data, achieving efficient knowledge transfer and mitigating catastrophic forgetting. The two tuning principles proposed in the paper, “expansion” and “compression,” along with their corresponding implementation schemes, demonstrate innovation and practical value.

**Strengths:**

1. The paper presents a prompt-based continuous spatio-temporal forecasting framework, EAC, introducing the “expansion” and “compression” principles and offering a new perspective on solving dynamic streaming spatio-temporal data prediction problems.
2. EAC can be combined with different STGNN architectures and performs well on various spatio-temporal data types.
3. By freezing the base STGNN model and adjusting a limited number of parameters in the prompt parameter pool, EAC can improve speed and reduce the number of parameters to be adjusted, demonstrating its efficiency.

**Weaknesses:**

Overall, my concerns are mainly about experiments.
(1) How does the performance of the schema adopt all historical spatio-temporal data for training, which is not mentioned in Fig. 1? It would be better if the performance of such schema were also discussed and included in the performance comparison.
(2) Section 5.2 provides a detailed comparison between different methods, and a further discussion on the difference in results across different domains (weather, traffic, and energy) should also be provided.
(3) The efficiency of EAC is observed to be largely influenced by the scale of the dataset in Section 5.4. Thus, a more in-depth analysis of the impact of the dataset scale on the model performance should be provided. This makes the real-world application questionable.
(4) Many details of the baselines and datasets are missing.
(5) More baselines published in 2023 and 2024 should be considered.

**Questions:**

Please address the questions above.

**Details Of Ethics Concerns:**

None.

---

> ### Author Response · Authors · 2024-11-15
> **The First Part of the Response to Reviewer S8oR**
>
> Dear Reviewer S8oR,
>
> We greatly appreciate your hard work during the review process. We are also pleased that you recognized several key features of our work in the Summary and Strengths section: _**innovation**_ and _**practical value**_, offering a _**new perspective**_, _**universality**_ (applicable to various STGNNs), _**acceleration of training**_, and _**reduction in the number of adjustable parameters**_. At the same time, we fully understand your concerns regarding the experimental section and are thankful for your detailed and insightful feedback. We believe **all of these are misunderstandings or omissions of information**. We will address your concerns **one by one** to clarify any confusion. Specifically:
>
> ---
>
> > **Q1: Why not use all historical spatio-temporal data for training? Could you try including results using this approach?**
>
> - We are happy to clarify this point. Stacking all historical data together for training might seem reasonable at first glance, but it is actually impractical and reflects a misunderstanding of continuous spatio-temporal graph learning. As we pointed out in the last sentence of the first paragraph of the introduction: *"Due to computational and storage costs, it is often impractical to store all data and retrain the entire STGNN model from scratch for each time period."* Thus, the primary motivation behind continuous spatio-temporal graph modeling methods is:
>
>   - **(Training and Storage Costs):** Storing all historical data and retraining is associated with unacceptable training and storage costs. Training costs are easy to understand, but storage costs are significant because the model is usually only a fraction of the size of the data (e.g., in the PEMS-Stream benchmark, the _**model size**_ per year is  _**36KB**_ compared to the _**dataset size**_ of _**1.3GB**_, approximately _**37,865:1**_).
>
>   - In addition to this fundamental motivation, we would like to share further insights:
>
>     - **(Privacy Risks):** In common continuous modeling tasks such as vision and text, a key improvement direction is to avoid accessing historical data, as this poses privacy risks beyond storage costs [1]. Accessing models that store knowledge from historical data is clearly safer. Common improvements, such as regularization-based and prototype-based methods, are moving in this direction.
>
>     - **(Practical Impossibility):** Unlike vision and text tasks, spatio-temporal graphs have a unique property: their nodes are constantly changing. This introduces practical issues that make it nearly impossible to implement. For example, when training a neural network, _**data must be fixed into a certain format for each batch**_ to fully leverage GPU batch processing capabilities. The number of nodes in spatio-temporal graphs changes across different periods, _**making this impractical**_. Therefore, most methods seek a backbone STGNN independent of node count to accept spatio-temporal graph data from different time periods, but it still requires that node counts must be consistent during training within the same period.
>     - **(Existing Approximation Methods):** The existing Online-ST methods can be seen as an approximation solution to training with all historical data. However, this often suffers from catastrophic forgetting and the need for full parameter adjustments, issues that our EAC effectively addresses.
>
>   - We sincerely hope these insights will help clear up any misunderstandings.
>
> [1] Wang, et al. "A comprehensive survey of continual learning: theory, method and application." *IEEE TPAMI,* 2024.
>
> ---
>
> >**Q2: Provide more discussion on the differences in results across different domains (datasets).**
>
> - We are happy to provide further analysis. Specifically, the primary difference between the three spatio-temporal datasets lies in the underlying spatio-temporal dynamics, or spatial dependencies:
>   - **(Differences in Underlying Spatio-temporal Dynamics):** In order, *Energy-Stream* involves a small wind farm with closely spaced turbines that share similar spatial patterns. *PEMS-Stream* represents the entire transportation system in Southern California, with moderate spatial dependencies. *Air-Stream* goes further, encompassing air quality records from air monitoring stations across all of China, leading to much more complex spatial dependencies. These differences are reflected in the MAE and RMSE metrics, with the performance progressively degrading as the complexity increases.
>   - **(Special Characteristics of Wind Farm Data):** Regarding the MAPE metric, we must note that in the *Energy-Stream* dataset, turbines occasionally face overheating protection or shutdown for inspection, which can cause some devices to produce very small values during certain time periods. This causes large percentage errors when using MAPE.

---

> ### Author Response · Authors · 2024-11-15
> **The Second Part of the Response to Reviewer S8oR**
>
> ---
>
> >**Q3: Would EAC be affected by dataset size? Please provide deeper analysis of the impact of data size on model performance and the potential risks in practice.**
>
> - In addition to the analysis mentioned in the paper, we are happy to provide more detailed insights to alleviate your concerns. Specifically:
>   - **(Adjustable Parameter Count):** As we introduced in Section 4.2, there is no free lunch in this regard. We acknowledge that while introducing the prompt parameter pool significantly improves performance, there is an inevitable risk of parameter bloat as the dataset size grows with more nodes. Therefore, we dedicated an entire section to empirical observations and theoretical analysis to explain how to reasonably eliminate redundant parameters, offering a _**compress principle**_. Consequently, the adjustable parameter count can be dynamically reduced with changes in _**k**_. In the hyperparameter analysis section, we further demonstrate that even with a limited number of tuning parameters, our approach still achieves SOTA performance. Thus, in large-scale scenarios, we can appropriately select the hyperparameter _**k**_ to balance performance and efficiency.
>   - **(Freezing Backbone Model):** One of the main advantages of EAC is that, compared to existing methods, freezing the backbone model directly leads to significant efficiency improvements, even in large-scale datasets. For example, in Figure 6, we maintained the fastest average training speed on the *air-stream* dataset, and this can be further accelerated by adjusting parameter k. Therefore, our approach is clearly the best choice compared to others.
>   - **(Advantages of In-Memory Storage):** Another point worth noting is that we need to point out that our prompt parameter pool is separate from the backbone model. Therefore, we can naturally store it in memory and only load it when needed. Therefore, for practical applications, overloading the prompt parameter pool is unnecessary worry.
>   - **(New Largest-Scale Benchmark):** We also would like to point out that the current mainstream benchmark for continuous spatio-temporal learning is *PEMS-Stream*, which has over 800 nodes. In this paper, we further gathered and constructed benchmark datasets from various domains (including meteorology and energy) and different scales (with more and fewer nodes), aiming to provide a richer evaluation for future work. Notably, *Air-Stream* includes spatio-temporal data from air monitoring stations across China, which we believe is practical for deployment. As for even larger global datasets, the current backbone model size would clearly be insufficient and would need to be scaled up, as seen in current large-scale spatio-temporal models [2, 3]. In contrast, the growth of the prompt parameter pool can be considered a more manageable solution.
>
> [2] Lam, et al. "GraphCast: Learning skillful medium-range global weather forecasting." *Science,*, 2023.
>
> [3] Shi, et al. "Time-MoE: Billion-Scale Time Series Foundation Models with Mixture of Experts." *arXiv,* 2024.
>
> ---
>
> >**Q4: Missing details on baselines and datasets.**
>
> - A friendly reminder: it appears that you missed the appendix materials, which fully address your concerns regarding baseline and dataset details. However, we will briefly summarize and respond to these issues here for clarity:
>   - **(Baseline Details):** In Section 5.1, we thoroughly discuss the specific approaches for various baselines and categorize several advanced methods. We also provide detailed descriptions, code links, and parameter settings for these methods in Appendix C.2. Notably, we also offer an anonymous repository containing all experimental results, training logs, and model weights of all baselines.
>   - **(Dataset Details):** In Section 5.1, we present the detailed information about all datasets and experimental setups. Further, in Appendix C.1, we provide additional details on dataset construction, feature selection, and statistical information.
>
> We believe we have provided sufficient baseline and dataset details. If you need any additional specifics, please feel free to provide more precise requests.

---

> ### Author Response · Authors · 2024-11-15
> **The Third Part of the Response to Reviewer S8oR**
>
> ---
>
> **Q5: Should more recent baselines from 2023 and 2024 be considered?**
>
> - **(Comprehensive Survey):** We are genuinely puzzled by this comment, as we have thoroughly surveyed all relevant papers in this field and compared all comparable baselines. Specifically, three of the four advanced improvements we included in our work [4, 5, 6] are from 2023 and 2024. In the Related Work section, we summarized other works that are not directly comparable but still relevant. We have also provided detailed responses to Reviewer 7vMi and Reviewer vM4w regarding why some works are not comparable. If you have specific baselines in mind, please do let us know.
> - **(Fair Comparison):** Another important detail is that we conducted fair comparisons for all approaches, including unified parameter settings and multiple rounds of experimentation. In particular, all experiments in this paper were repeated five times with random seeds to ensure fairness. For example, for the seven baselines across three different datasets, we performed a total of 7*(4+7+4)*5=525 experiments, which incurs substantial research costs that are often overlooked.
> - **(Further Performance Improvement):** Lastly, although we do not emphasize it, Table 1 results are not the upper limit of our approach. For example, by changing the backbone network, we can further improve performance, as shown in Table 3. We believe this benefit is not achieved by other methods.
>
> [4] Wang, et al. "Pattern expansion and consolidation on evolving graphs for continual traffic prediction." *SIGKDD,* **2023**.
>
> [5] Wang, et al. "Knowledge expansion and consolidation for continual traffic prediction with expanding graphs." *IEEE TITS,* **2023**.
>
> [6] Lee, et al. "Continual Traffic Forecasting via Mixture of Experts.", *arXiv* **2024.**
>
> ---
>
> In summary, we believe our approach offers a rich exploration and provides a new option for continuous spatio-temporal graph learning tasks. Given its lightweight, efficient, effective, and universality characteristics, it has the potential to serve as a new suitable baseline. We hope the responses above address your concerns, and if possible, **we kindly request that you reconsider increasing the score**. Should you have any further suggestions, we are more than happy to discuss them and make necessary improvements to the paper.
>
> Cheers,
>
> All authors

---

> ### Author Response · Authors · 2024-11-21
> **Kindly Request for Reviewer's Feedback (Before the Rebuttal Deadline!)**
>
> Dear Reviewer S8oR,
>
> **Since the End of the Rebuttal is coming very soon - only a few days left, we would like to inquire if our response addresses your primary concerns.** If it does, we kindly request that you reconsider the score. If you have any additional suggestions, we are more than willing to engage in further discussions and make necessary improvements to the paper.
>
> Thanks again for dedicating your time to enhancing our paper!
>
> Looking forward to your feedback.
>
> Best,
>
> Authors

---

### Author Response · Authors · 2024-11-21
**General Response to All Reviewers**

We are truly grateful for the invaluable time and detailed feedback provided by all the reviewers. _**It is encouraging to see that almost every reviewer has recognized the positive aspects of our manuscript**_, such as `important and practical problem` (Reviewers S8oR, vM4w, YV6D), `strong and reasonable motivation` (Reviewers S8oR, 7vMi, vM4w, YV6D), `novel perspective and method` (Reviewers S8oR, 7vMi, vM4w, YV6D), `solid empirical and theoretical analyses` (Reviewers 7vMi, vM4w, YV6D), and `impressive results` (Reviewers S8oR, 7vMi, vM4w, YV6D).

We have provided detailed responses to each reviewer’s feedback. In this general response, we outline the major revisions made to our `new manuscript` based on the valuable suggestions provided by the reviewers. **(Please check the changes in purple font in our revision)** We hope these responses adequately address any potential concerns from the reviewers.

+ _**Presentation**_: Based on the feedback from **Reviewer YV6D**, our revisions include ① modifying the introduction part of Section 4 to more clearly emphasize the learning process of the prompt parameter pool, and ② updating Figure 2 to enhance understanding.

+ _**Experiments**_: According to the feedback from **Reviewers 7vMi** and **YV6D**, our revisions include ① presenting results and analysis of experiments for each individual period in PEMS-Stream datasets (see Appendix D) and ② including comparisons of the results with the ST-CRL baseline (see Appendix C.2).

+ _**Discussion**_: According to the feedback from **Reviewers S8oR, 7vMi, and vM4w**, our revisions include ① adding more discussion on parameter expansion (see Appendix F.1), ② including discussions about using all historical data (see Appendix F.1), ③ providing a discussion on performance differences across different domains (datasets) (see Appendix C.1), and ④ adding more details on the differences between related works (see Appendix C.2).

The expertise of all reviewers has greatly helped us strengthen our manuscript! We have made sincere efforts to address all the issues raised and are deeply grateful for the recognition and suggestions from all reviewers. We still respectfully welcome further discussions from all reviewers.

Best regards,

All Authors

---

### Comment · Area_Chair_yQgw · 2024-11-25
**Acknowledge the author responses**

Dear Reviewers,

Thank you very much for your effort. As the discussion period is coming to an end, please acknowledge the author responses and adjust the rating if necessary.

Sincerely,
AC

---

### Comment · Area_Chair_yQgw · 2024-11-28
**Discussion needed**

Dear Reviewers,

As you are aware, the discussion period has been extended until December 2. Therefore, I strongly urge you to participate in the discussion as soon as possible if you have not yet had the opportunity to read the authors' response and engage in a discussion with them. Thank you very much.

Sincerely,
Area Chair

---

### Meta-Review · Area_Chair_yQgw · 2024-12-18

**Metareview:**

This paper proposes a prompt tuning-based continuous forecasting method, EAC, following two fundamental tuning principles guided by empirical and theoretical analysis.  Overall, the reviewers liked the motivation, rationale, method design, and evaluation results.  Some reviewers raised several concerns about evaluation setting, but the authors succesfully addressed such concerns during the discussion period.  Therefore, I recommend an accept.

**Additional Comments On Reviewer Discussion:**

* I disregarded the history data comment raised by Reviewer S8oR, because it seems to be wrong.
* I disregarded the baseline comment raised by Reviewer S8oR, because it seems to be wrong.
* The authors provided additional evaluation results, and Reviewer YV6D increased his/her rating.

---

### Decision · Program_Chairs · 2025-01-22

Accept (Poster)